# Optically detected magnetic resonance with an open source platform

Hossein Babashah,[1, 2, a)] Hoda Shirzad,[1, 3, a)] Elena Losero,[1, b)] Valentin Goblot,[1, 3] Christophe Galland,[1, 3] and Mayeul Chipaux[1, 3, c)]

[1)]*Institute of Physics, École Polytechnique Fédérale de Lausanne (EPFL), Lausanne CH-1015, Switzerland*
[2)]*WAINVAM-E, 1 rue Galilée, 56270 Plœmeur, France*
[3)]*Center for Quantum Science and Engineering, EPFL, Lausanne, Switzerland*

(Dated: 12 July 2023)

Localized electronic spins in solid-state environments form versatile and robust platforms for quantum sensing, metrology and quantum information processing. With optically detected magnetic resonance (ODMR), it is possible to prepare and readout highly coherent spin systems, up to room temperature, with orders of magnitude enhanced sensitivities and spatial resolutions compared to induction-based techniques, allowing for single spin manipulations. While ODMR was first observed in organic molecules, many other systems have since then been identified. Among them is the nitrogen-vacancy (NV) center in diamond, which is used both as a nanoscale quantum sensor for external fields and as a spin qubit. Other systems permitting ODMR are rare earth ions used as quantum memories and many other color centers trapped in bulk or 2-dimensional host materials. In order to allow the broadest possible community of researchers and engineers to investigate and develop novel ODMR-based materials and applications, we review here the setting up of ODMR experiments using commercially available hardware. We also present in detail the dedicated collaborative open-source interface named Qudi and describe the features we added to speed-up data acquisition, relax instrument requirements and extend its applicability to ensemble measurements. Covering both hardware and software development, this article aims to overview the setting of ODMR experiments and provide an efficient, portable and collaborative interface to implement innovative experiments to optimize the development time of ODMR experiments for scientists of any backgrounds.

## CONTENTS

[a)]H.S and H.B. contributed equally to this work.
[b)]Current affiliation: Division of Quantum Metrology and Nanotechnologies, Istituto Nazionale di Ricerca Metrologica (INRiM), Strada delle Cacce 91, 10135 Torino, Italy
[c)]Correspondence: mayeul.chipaux@epfl.ch

## INTRODUCTION

Spin is a quantum property of elementary particles which confers them an intrinsic magnetic moment that can be associated with a circulating charge flow in their wave function[1]. In atoms and ions, while the nuclear spins remain essentially shielded, the unpaired electrons result in magnetic moments that are three order of magnitude stronger – of the order of the Bohr magneton. It makes electron spins the main contributors to most materials' magnetic properties. In the modern context of the second quantum revolution[2], electron spins, such as in quantum dots[3,4], color centers in diamond[5] or silicon carbide[6], or rare earth ions[7] have been either studied as individual quantum systems or as ensembles. While electron spins naturally interact with various external and internal fields, they can also be isolated from unwanted perturbations by experimental design or choice of system[8], making them either great quantum sensors[9] or highly coherent quantum bits (or qubits) for quantum information processing[10,11].

Standard inductive radio- and microwave techniques for spin resonance measurements suffer from poor sensitivity and low spatial resolution. Indeed, under achievable magnetic fields electronic spin transition energies typically lie in the microwave (MW) domain, with associated Boltzmann temperatures of few tens of millikelvin, so that they are very weakly polarized at ambient conditions. Additionally, sensing MW fields at the single photon level remains a tremendous challenge[12] and the associated wavelengths (millimeter to centimeter) is by far larger than the few tens of nanometers required for directly probing spin-spin interactions[13]. Even using magnetic field gradients magnetic resonance imaging (MRI) cannot achieve spatial resolution below tens of micrometers[14]. All these limitations affect both the spatial resolution in quantum sensing and the scaling of entangled qubits' networks for useful quantum computing.

Optically detected magnetic resonance (ODRM)[15] offers a powerful solution to these problems. It is applicable to a range of material systems (see Table I) whose joint optical and magnetic properties render their electron-spin magnetic transition detectable in the optical domain. The following ingredients are characteristics of ODMR systems:

- Firstly, the electronic spin can be polarized by optical pumping to a much higher degree than under thermal equilibrium. Associated with naturally low spin-lattice coupling[16], certain electronic spins can maintain long polarization and coherence times even at elevated temperature[17]. Together these properties can totally relax the need for cryogenic conditions.

- Secondly, the strength of optical signal (typically absorption, fluorescence or phosphorescence) must depend on the electronic spin state or its projection along a quantization axis. Since optical fields can routinely be detected at the shot-noise limit, the sensitivity of magnetic resonance spectroscopy and the fidelity of spin state readout can be significantly increased while spin initialization and readout can be achieved with a much

higher spatial resolution set by the optical wavelength (or below).

- Finally, these advantages can be translated to nuclear spin spectroscopy through the use of hyperfine coupling[18].

Together, these features are placing ODMR at the core of many emerging spin-based quantum technologies.

Despite the growing number of researchers and engineers active in this area and the availability of advanced optical and microwave (MW) equipment, building and operating an ODMR setup for a specific purpose usually demands substantial hardware and software developments. Moreover, beyond the apparent simplicity of ODMR measurements, a certain amount of tacit knowledge is needed to avoid some pitfalls that can spoil measurement accuracy and reproducibility.

The object of this article is twofold: (i) to describe a typical ODMR setup, from its conception and construction to the optimization of specific measurements and (ii) to present, in details, the ODMR-dedicated open source platform *Qudi*[19] with the features we developed for speeding-up acquisitions, relaxing instrumental requirements, widening its applications and lowering entry barriers.

In Sec. I, we first recall ODMR main principles and measurement techniques and review the most studied material systems with their applications. We then present the hardware configuration needed for ODMR experiments in Sec. II. Aspects related to optical setup, generation of MW signals, pulses and bias magnetic fields are discussed. We show how some ODMR experiments (in continuous wave and pulsed modes) can be visualized directly with an oscilloscope and determine when the use of a computer interface is useful or required. In Sec. III, we describe how these instruments are connected together and present the most common strategies for synchronizing ODMR experiments. Both the computer and oscilloscope interfaces are covered. The section IV is dedicated to the open source platform *Qudi*, first release in 2017[19] and upgraded by us for efficient and versatile ODMR experiment operations. Finally, in Sec. V we give a few practical advice to perform reliable and contrasted measurements, in particular accounting for technical noise and instrument imperfections. We take a specific focus on NV center ensembles as an illustrative scenario.

We believe that the software experimental developments presented here will help newcomers achieve faster and better results for their desired application and provide experts an efficient, portable, and collaborative platform to explore innovative experiments.

## I. OPTICALLY DETECTED MAGNETIC RESONANCE

### A. Main principles

ODMR relies on two main features: (i) initialization (*i.e.* polarization) of the spin state under optical excitation (spin pumping) and (ii) spin-state-dependent optical properties,

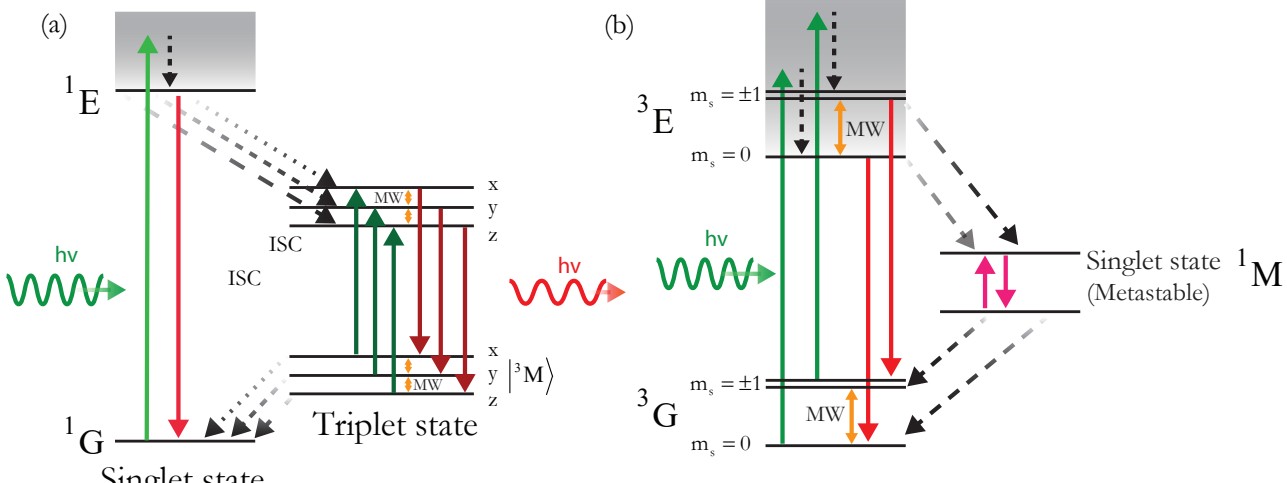

FIG. 1. InterSystem Crossing (ISC) based ODMR dynamics when the ground state is singlet (a) or triplet (b).

such as fluorescence, phosphorescence, absorption, or, as more recently demonstrated, photocurrent[20], used for spin state readout. While feature (i) boosts the contrast of the spin resonance signal, feature (ii) allows to advantageously trade low energy MW photons for much higher energy optical ones in the spin state measurement. These properties may arise from spin-orbit coupling (see Fig. 1), a relativistic effect that allows for inter-system crossing (ISC), *e.g.* transitions between singlet $S = 0$ and triplet $S = 1$ state manifolds[21,22] (in a spin-1 system). The probability of ISC may depend on the initial spin state, which in turn may result in spin-dependent optical properties and spin-dependent relaxation pathways. In other systems (see Sec. I C), ODMR can be achieved under resonant optical pumping such as using $\Lambda$-shaped configurations.

**B. Interactions with external quantities**

In a system that is not isotropic, a quantization axis emerges; noting $m_S$ the spin quantum number (*i.e.* the spin projection along that axis), the dipolar spin-spin interaction lifts the degeneracy between the states $|m_S = 0\rangle$ and $|m_S = \pm 1\rangle$ among the $S = 1$ triplet, by an amount called zero-field splitting (ZFS), often denoted with $D$. Further degeneracy is lifted between $|m_S = -1\rangle$ and $|m_S = +1\rangle$ by a parameter $E$ when the symmetry is less than axial. The eigenstates, potentially made of superpositions of pure spin states, are then often denoted $x$, $y$ and $z$. $D$ and $E$ depend on geometrical aspects and therefore on pressure and temperature. In addition, Stark and Zeeman effects further separate the $|m_S = -1\rangle$ and $|m_S = +1\rangle$ state which makes ODMR systems naturally sensitive to magnetic and electric fields. Finally, further interactions may occur with other electron or nuclear spins, resulting in more level splittings, spin dephasing and diffusion, etc.

**C. Main ODMR systems and their applications**

We can distinguish two main configurations linked to the multiplicity of the ground state (see Fig. 1). The ground state of organic molecules are typically spin singlets, where all electrons are paired in covalent bonds or lone pairs[23]. Photoexcitation followed by ISC may promote the molecule into a spin triplet state and enable ODMR as in Fig. 1 (a). Since 1967[24–26], such process has been exploited to enhance the detection limit of magnetic resonance[23] with a particular application in photosynthesis research[27,28]. ODMR from a single molecule was reported for the first time in 1993 independently by Wrachtrup *et al.*[29] and Köhler *et al.*[30]. Both experiments relied on the photoexcited metastable triplet state of a pentacene molecule trapped in a p-terphenyl host crystal. Pentacene's spin-dependent optical properties have also been used inside MW resonators[31], such as for demonstrating the first room temperature maser[32] or conversely for cooling a microwave mode[33].

The negatively charged Nitrogen-Vacancy center ($NV^-$) in diamond - hereafter called NV center - belongs to the second configuration depicted in Fig. 1 (b). Its ground $^3A_2$ and main excited $^3E$ optical states are spin triplets. The first single NV center ODMR signal was reported by Gruber *et al.* in 1997[34]. For the following reasons, NV centers then became the most studied ODMR system:

- ISC combined with a low Debye-Waller factor allows for an efficient off-resonance optical pumping and readout over a large range of wavelengths, even far over room temperature[35] and under high pressures[36].

- Diamond is bio-compatible making NV-doped nanodiamonds particularly relevant[37] when used as biomarkers[38–40] or quantum sensors[41–43].

- NV centers can exhibit long spin relaxation and coherence times[17] thanks to the low density of nuclear

spins in diamond and the decoupling of spin states from lattice phonons[16]. It makes them versatile quantum sensors[44] for detection and imaging of magnetic and electric fields, temperature, strain, currents and associated noises[9,45,46], as well as for microwave field imaging[47] or spectroscopy[48,49].

- NV center ODMR properties can be extended to polarize[50–52] and/or detect[53–56] other electron or nuclear spins in their vicinity and perform their magnetic resonance spectroscopy[57,58]. In these cases the sensitivity can go down to a single nuclear spin, orders of magnitude better than conventional electron paramagnetic resonance (EPR) and nuclear magnetic resonance (NMR) methods.

- Finally, thanks to their excellent coherence properties and couplings to even longer lived nuclear spins[59], NV centers are also promising qubits for quantum information processing[11] and fundamental studies of quantum mechanics[60].

Other diamond defects from the group IV-vacancy category[61] have recently revealed their ODMR signals, such as the negatively charged silicon-vacancy center (SiV$^-$)[62] and its neutral form (SiV$^0$)[63], the Germanium-Vacancy (GeV)[64] and the Tin-Vacancy (SnV)[65] centers. In these cases, ODMR is possible only at cryogenic temperatures since it requires resonant optical addressing – differing from the scheme presented in Fig. 1. Single-defect ODMR signals have recently been reported in silicon carbide[6,66,67] or hexagonal boron nitride[68,69]. Notably, the so called V1 center in silicon carbide[66] involves a spin quartet $S = \frac{3}{2}$ ground state. Rare earth ions in crystal lattices are also intensely studied for long lasting and broadband optical quantum memories[70,71]. One can cite $Ce^{3+}$ in YAG[72] or $Yb^{3+}$[73] and $Eu^{3+}$[74] in $Y_2SiO_5$. Semiconductor quantum dots have also been used in ODMR experiments[75,76]. The optical wavelengths and microwave frequencies associated to those systems are summarized in Table I.

## D. Main ODMR techniques

### 1. Continuous wave mode

ODMR can be performed under continuous wave (CW) optical and MW excitation. A CW-ODMR spectrum is obtained by recording the optical emission or absorption as function of the MW frequency that is being swept. When spin resonances are reached, dips or peaks appear in the recorded optical intensity (see Fig. 2), allowing to measure physical parameters affecting the resonance frequencies or to determine the gyromagnetic ratio of the system itself (EPR spectroscopy). For a given system, several external parameters affect the precision of the spectrum obtained by CW-ODMR through the change in signal-to-noise ratio and spin resonance linewidth. These parameters include MW power (via power broadening[80] and sample heating), laser power (via change

in excitation rate, governing spin polarization, optical emission and photo-ionization rates), MW sweeping method and pace, etc. Optimization of CW-ODMR signal acquisition is discussed in Sec. V B.

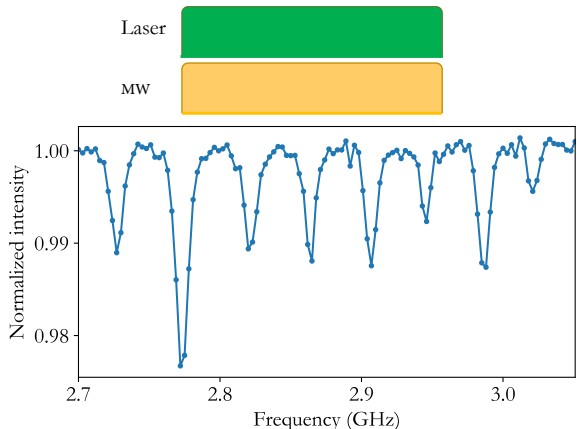

FIG. 2. Example of a continuous-wave ODMR spectrum from an NV-center ensemble under d.c. magnetic field, displaying eight resonances (hyperfine splitting not resolved) with contrasts on the order of 1%.

### 2. Pulse sequences

Pulsed ODMR has the first advantage of mitigating power broadening by letting the system evolve freely after optical and MW excitation. It also allows to tune the system for being specifically sensitive to certain quantities and insensitive to unwanted perturbations, taking advantage of decades of development in the field of magnetic resonance spectroscopy[81]. Good practices for optimizing a pulsed ODMR signal are discussed in Sec. V C. Below, we first introduce the most elementary pulse sequences (in the case of a spin triplet).

Common to most sequences, a first laser pulse initializes the system (e.g. in state $|m_S = 0\rangle$). After a period of spin manipulation with MW pulses and/or free evolution, a second laser pulse interrogates the spin state projection along the $(|m_S = 0\rangle, |m_S = +1\rangle)$ basis through the optical emission or absorption intensity. It also re-initializes the system for the subsequent measurement. In the Bloch sphere representation, a MW $\pi$-pulse performs a $\pi$ rotation along a certain axis; for instance, it can swap the $|m_S = 0\rangle$ and $|m_S = +1\rangle$ (or $|m_S = -1\rangle$). A $\pi/2$ pulse only performs a quarter turn, placing an initial $|m_S = 0\rangle$ state into an equal superposition $\frac{1}{\sqrt{2}}\left(|m_S = 0\rangle + e^{i\phi}|m_S = +1\rangle\right)$ (or $|-1\rangle$).

*Relaxometry – $T_1$ measurement* — This sequence, reported in Fig. 3(a), requires no MW fields. The two initialization and readout optical pulses are separated by a varying dark time $\tau$. It allows for probing the relaxation dynamics from an initialized pure state to a thermal equilibrium. Interesting for applications, the relaxation time, or $T_1$, can sense the magnetic noise of the local environment[82,83] such as the

| System (host) | Optical pumping [Off-resonance] (On-resonance (ZPL)) | Microwave Zero field splitting $D$ and $E$ Gyromagnetic ratio $\gamma$ | Comments / Applications |
|---|---|---|---|
| **Organic molecules** | Various | $D = 500$ to $1500\,\text{MHz}$ $E = 0$ to $500\,\text{MHz}$ | Sensitive EPR spectroscopy[23] Photosynthesis[77] |
| **Pentacene** (p-terphenyl crystal) | [Green - yellow, *e.g.* 585 nm] | $D = 1395\,\text{MHz}$ $E = 53\,\text{MHz}$ | First single molecule ODMR signal[29,30] First room temperature maser[31,32] |
| **NV$^-$ center** (Diamond) | [Green *e.g.* 515 or 532 nm] (637 nm = 1.945 eV) | $D = 2.87\,\text{GHz}$ $\gamma = 28\,\text{GHz}\,\text{T}^{-1}$ | Quantum sensing Quantum computation |
| **Group IV-Vacancy centers** (Diamond) | | | Cryogenic temperature.[78] Quantum networks[79] |
| **SiV$^-$** | (737 nm) | $D = 50\,\text{GHz}$ | [62] |
| **SiV$^0$** | (946 nm) | $D = 944\,\text{GHz}$ | [63] |
| **GeV** | (612 nm) | $D = 170\,\text{GHz}$ | [64] |
| **SnV** | (260 nm) | | [65] |
| **V1 center** | (861 nm) | $D = 2\,\text{MHz}$ | Spin quartet[66] |
| **PL8 center** (4H-Silicon carbide) | [920 nm] (1007 to 1024 nm) | $D = 1.39\,\text{GHz}$ $E = 4\,\text{MHz}$ | Room temperature[67] [6] |
| **Unknown defects** (Boron Nitride) | [532 nm] | $D = 0.1$ to $2.4\,\text{GHz}$ | [68] |
| **Rare earth ions** | | | Quantum memories[70,71] |
| $Ce^{3+}$ (YAG) | (460 and 486 nm) | $< 22.2\,\text{GHz}$ | [72] |
| $Yb^{3+}$ ($Y_2SiO_5$) | (980 nm) | $< 2.62\,\text{GHz}$ | [73] |
| $Eu^{3+}$ ($Y_2SiO_5$) | (580 nm) | 34.5 and 46.2 MHz | [74] |
| **Quantum dots** (In,Al)As/(AlAs) | [360 and 404 nm] | | [75] |
| CdSe/(Cd,Mn)S | [405 nm] | 60 GHz | [76] |

TABLE I. Summary of some popular ODMR systems and their main parameters.

one created by free radicals species in living cells[41,42] or by chemical reactions[84].

*Rabi sequence – $\pi$-pulse calibration* — In this sequence, reported in Fig. 3(b), the delay between the optical pulses is fixed and a MW pulse is applied for a varying duration $\tau$ (alternatively, its power can be varied). The measured optical signal as a function of $\tau$ shows Rabi oscillations. From this measurement the duration of $\pi$ and $\frac{\pi}{2}$-pulses can be inferred as half and quarter periods of the oscillations, respectively.

*Ramsey sequence – $T_2^*$ measurement* — This sequence, reported in Fig. 3(c), consists in a pump-probe experiment where two $\pi/2$ pulses are inserted. The first $\pi/2$, immediately after the initialization laser pulse, induces spin precession. After the free evolution time $\tau$ the precession is halted by a second $\pi/2$ pulse before the read-out laser pulse is applied. The optical signal as function of $\tau$ presents oscillations at the difference-frequency between the applied MW field and the spin transition being probed. Therefore, this sequence can be used for d.c. sensing and offers better sensitivity compared to the CW-ODMR approach[80] by eluding the detrimental effect of power broadening. Ramsey fringes are damped with a time constant reflecting the spin dephasing time or heterogeneous decoherence time called $T_2^*$.

*Hahn echo sequence – $T_2$ measurement* — This sequence, reported in Fig. 3(d), uses a $\pi$-pulse in the middle of the Ram-

sey sequence. This approach mitigates the effect of slowly fluctuating fields (such as from the nuclear spin bath) by canceling in the second half of the evolution the accumulated phase difference from the first half and restoring the original coherence. The characteristic decay time measured with this protocol is usually called $T_2$, and can typically be one or two orders of magnitude longer than $T_2^*$. More and more complex and faster dynamical decoupling sequences can be employed to extend $T_2$; yet, it remains bounded by relaxation time through $T_2 < 2 \cdot T_1$. Such echo sequence is particularly suitable of a.c. field sensing as it can filter a narrow frequency window[9].

Many efforts have been devoted to develop and optimize pulse sequences for increasing sensitivities to specific quantities and frequencies while canceling other noises. A comprehensive review dedicated to NV centers can be found in[44].

## II. HARDWARE FOR CONTINUOUS AND PULSED ODMR

We review here the instrumental requirements for ODMR measurements in terms of optical setup, MW generation and biased magnetic field. Both CW and pulsed ODMR measure-

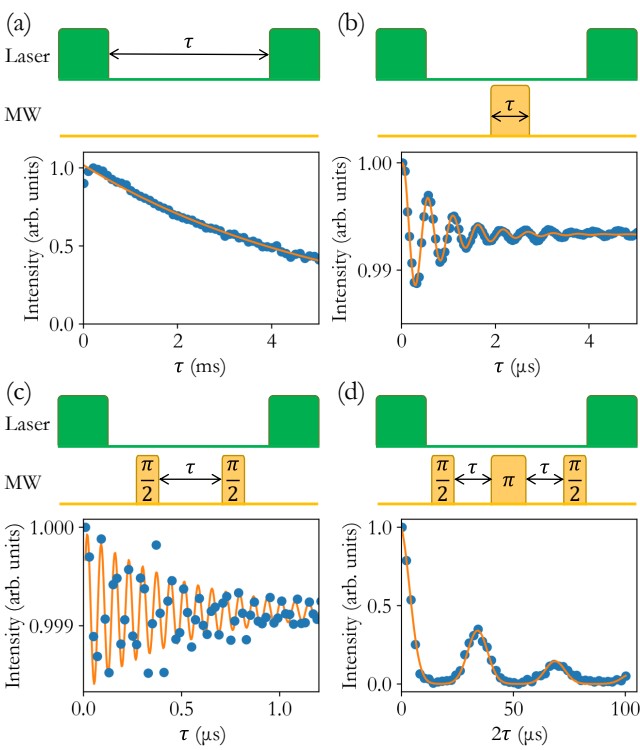

FIG. 3. Most common pulse sequences for ODMR: a) Relaxometry b) Rabi sequence c) Ramsey sequence d) Hahn echo sequence. In each panel, laser and MW pulses are represented schematically in the top part. In the bottom part, NV centers' experimental data obtained in our set-up from are shown.

ments are discussed. We take the diamond NV center as an accessible example (green pumping, red photoluminescence (PL)). However, the properties of other defects summarized in Table I can be consulted in order to adapt the equipment for their study, with a special care given on the optical wavelengths, microwave frequencies and possible need for cryogenic environment.

## A. Photoluminescence microscopy

As mentioned in Sec. I, ODMR relies on monitoring an optical signal such as the PL. It is obtained by directing a pumping beam (a laser or LED) onto the system of interest and routing the luminescence signal onto a photodectector (such as an amplified photodiode, a single photon counting module or a camera).

The simple optical setup reported in Fig. 4 analogous to an epifluorescence microscope can represent the minimal required optical setup to perform ODMR. An excitation laser beam is reflected on a dichroic mirror and focused on the sample via a microscope objective or a simple lens. The sample's luminescence is collected by the same lens and sent to a photodetector. Alternatively, the sample can be pumped from another side or by total internal reflection[85–87] to relax the need for a dichroic mirror or optimize pump power utilization and

spatial homogeneity of excitation. An inverted configuration of the microscope (*e.g.* where the objective is below the sample) is preferred when using liquid immersion objectives, especially for biological applications[88].

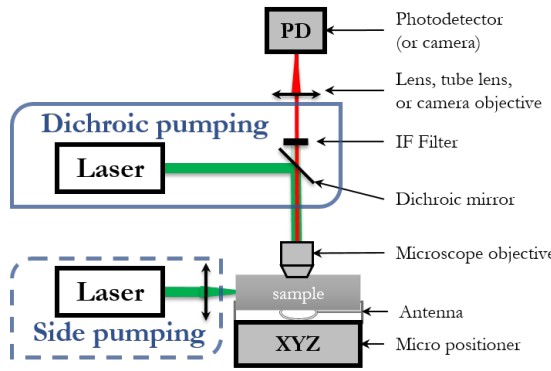

FIG. 4. Schematic of a simple OMDR setup.

### 1. Light detector and imaging mode

The choice of the light detector is probably the most impacting point in the development of a new setup. It is governed by two main aspects:

*Single-pixel vs. wide-field imaging*     — In the former case, a photodetector is sufficient, with appropriate filters placed in front of it to discriminate the signal of interest from the scattered laser light and the possible background luminescence from ODMR-inactive centers[89]. At the cost of small loss of transmitted power, spatially filtering the luminescence signal through a confocal pinhole or an optical fiber reduces the out-of-focus background and collection area (see Fig. 5). If imaging capability is desired it can be obtained by confocal scanning (see II A 5 and Fig. 5).

For wide-field imaging, a pixel array (camera) is required in combination with a large and homogeneous illumination area. Comprehensive reviews about wide-field imaging in the particular case of NV centers can be found in[90–92]. Besides, accordingly to the specific experiments, different cameras (*e.g.* CCD or a CMOS) can be preferred[93].

*Strength of optical signal*     — It depends on the number of luminescent centers or molecules being addressed, their individual quantum yields and the collection + detection efficiency. For the study of single or few emitters, a digital photon counter (d-PC) is necessary, such as a reverse-biased avalanche photodiode operated in Geiger mode or a superconducting nanowire detector. d-PCs generate electrical pulses (*e.g.* TTL) upon photon absorption with a quantum efficiency that can exceed 80%, allowing for the study of photon fluxes down to sub-kHz level, limited by the dark count rate of the detector.

For measurements on large ensembles, d-PCs typically saturate at count rates of few tens of MHz (*i.e.* few picowatts at visible wavelengths). Above that analog photodectors

(a-PDs) can be used, such as amplified or avalanche photo-diodes. In a wide-field configuration with a weakly emitting system, particularly sensitive cameras are required such as Electron Multiplying CCD (low noise) or phosphorus intensified camera (gated).

To enhance the sensitivity in under significant background signal, one can modulate and demodulate the ODMR signal (*e.g.* through the applied MW frequency) with the help of a lock-in amplifier[94]. Such methods can be of great help in the context of biosensing where background emission is strong[40]. For wide-field imaging, the recently developed arrays for lock-in detector[93] may find numerous applications in the upcoming years.

### 2. *Optical components and alignment*

When addressing individual emitters, one of the most important parameters is the objective's numerical aperture (*NA*) defined as $NA = n \sin\theta$, with $n$ the refractive index of the immersion medium and $\theta$ the half-cone angle of collection. For isotropic emission pattern, it limits the fraction of collected to emitted light to:

$$\eta < \frac{1}{2}(1 - \cos\theta) \tag{1}$$

with $\cos\theta = \sqrt{1 - (NA/n)^2}$.

Through diffraction, it also limits the achievable resolution $r_{min}$ defined as the minimum distance at which two objects can be distinguished. As commonly determined by the Rayleigh criterion:

$$r_{min} = \frac{r_{Airy}}{2} = 0.61 \cdot \frac{\lambda}{NA} \tag{2}$$

where $\lambda$ is the wavelength of the optical source and $r_{Airy}$ the radius of its diffraction pattern on the sample. Similarly, the axial (depth) resolution is set by:

$$z_{min} = 1.4 \frac{\lambda n}{NA^2} \tag{3}$$

A larger *NA* is usually obtained at the cost of a smaller working distance. As a good compromise, an air objective with *NA* up to 0.9 can offer a working distance of 1 mm. In contrast, in a confocal setting, the magnification $M_{obj}$ of the microscope objective bears no direct relation to spatial resolution and collection efficiency. It matters however for the design of the excitation and collection optics as well as in the case of wide-field imaging on a pixel array.

To achieve a spatial resolution limited by Abbe's diffraction and avoid unnecessary light loss, care should be taken to the following aspects:

- Firstly, the excitation beam diameter $\Phi$ should match or exceed the back aperture diameter of the microscope objective given by:

$$\Phi = 2 f'_{obj} \tan\theta \tag{4}$$

with $\tan\theta = \frac{NA/n}{\sqrt{1-(NA/n)^2}}$, $f'_{obj} = \frac{l_{tube}}{M_{obj}}$ the effective focal lens of the microscope objective, $l_{tube}$ being the tube lens as defined by the microscope supplier (*e.g.* 180 mm for Olympus, 200 mm for Nikon and Mitutoyo and 165 mm for Zeiss).

- Secondly, the confocal pinhole of radius $r_{hole}$ (or optical fiber of numerical aperture $NA_{fiber}$) has to be chosen to match the diffraction pattern of the photoluminescence beam on the pinhole (or fiber). When the image is sent to infinity after the objective the total magnification up to the pinhole (or fiber) equals:

$$M_{tot} = M_{obj} * M_{lens} = M_{obj} \frac{f'_{lens}}{l_{tube}} \tag{5}$$

where $f'_{lens}$ is the focal length of the lens used to focus on the pinhole. So to provide good confocal resolution while avoiding significant optical loss, $r_{hole}$ can be chosen to match the radius of the Airy diffraction pattern:

$$r_{hole} = 1.22 \times M_{tot} \frac{\lambda_{PL}}{NA} \tag{6}$$

Here, $\lambda_{PL}$ refers to the wavelength of the photoluminescent signal. The pinhole can also be chosen smaller. This would enhance the resolution (*e.g.* reduce $r_{min}$) by a factor down to $2/3$, at the cost of more optical loss.

- If an optical fiber is used instead, its numerical aperture can be taken as:

$$NA_{fiber} = \frac{NA}{M_{tot}} \tag{7}$$

In those cases, either the focusing lens, the pinhole or optical fiber can be changed to obtain the desired spatial filtering. Further details about the basics of confocal microscopy can be found in[95].

Many efforts have been devoted to surpass the diffraction limit in ODMR using super-resolution techniques such as stochastic optical reconstruction microscopy (STORM) and stimulated emission depletion (STED) microscopy[96,97].

### 3. *Light source*

For the most studied ODMR systems (Sec. I C), the optical pumping can be performed off resonance. Consequently, there is no stringent requirement on temporal coherence and linewidth of the light source, so that even a white-light LED can be employed, as long as suitable filter sets are used to prevent leakage of excitation light into the detector. However, fluctuations in laser power affects the contrast of the ODMR, as well as the stability and coherence of the NV centers. Therefore, it is important to carefully control the laser power fluctuation against temperature.

Transverse spatial coherence (single mode emission) is required in a confocal configuration in order to obtain a

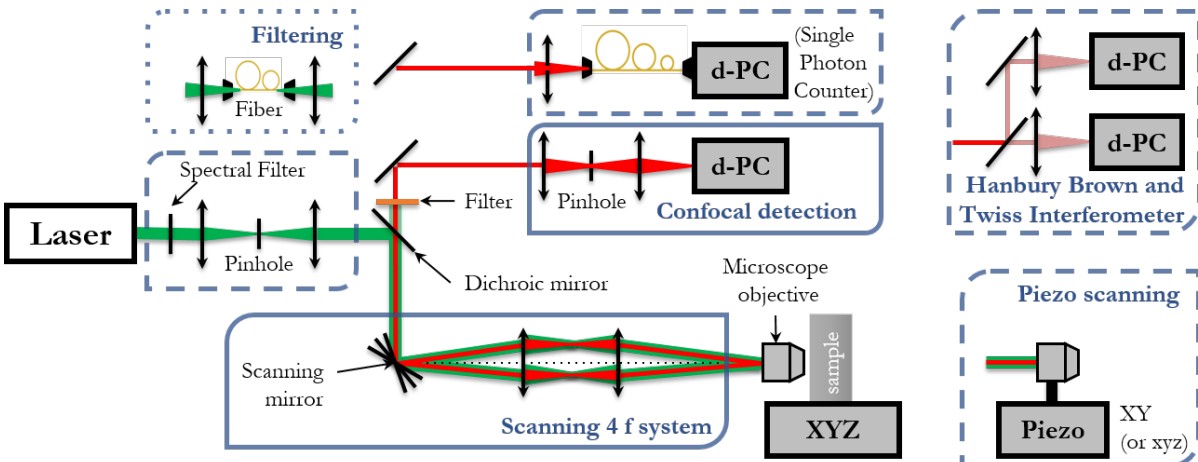

FIG. 5. Schematic of a typical home-build confocal microscope used for low light ODMR experiments. The collimated laser beam is spectrally filtered (to remove broadband spontaneous emission) before being reflected on the dichroic mirror and focused on the sample using a high-NA objective lens. Photoluminescence is collected by the same objective, passes through the dichroic mirror and an interference filter (band-pass or long-pass) to reject the excitation light, and is optionally spatially filtered using a confocal pinhole or an optical fiber before its intensity is measured by the detector. In order to scan the sample both a scanning mirror or a piezo stage can be used. To study single emitters, a Hanbury-Brown and Twiss interferometer can be added.

diffraction-limited spot size at the focus. As shown in Fig. 5, a spatial filter (pinhole or mono-mode fiber) can be introduced in the excitation path to improve transverse coherence. In this case, and using large NA, optical powers of milliwatts or less are often sufficient to saturate the emitter. Therefore, commercially available solid state lasers are a great and affordable option. Optionally, neutral density filters allow to adjust the power incident on the sample while maintaining the laser at its optimal operating power. In the wide-field configuration, the laser power is diluted over the entire area of interest, requiring high power lasers up to watts level. In all cases, temporal fluctuation in laser power represents the main source of technical noise, which can be mitigated by adding a feedback control loop or monitoring laser power in real time through a second detector. Other forms of technical noises related to pulsed measurements are discussed in Sec. II C.

### 4. PL digitization and visualization

Different approaches are used to measure the PL as a function of the type of signal coming from the photodetector.

Analog photodetectors (a-PDs) deliver an electric signal proportional to light intensity (temporally averaged over a given bandwidth) that can be visualized directly on an oscilloscope for both continuous and pulsed measurements. Alternatively, a computer interfaced acquisition card that embeds an Analog to Digital Converter (ADC) can be used. In this context, the most relevant features of these data acquisition cards (DAQ) are the number of input and output channels and the maximum sampling rate. The sampling rate sets an upper limit to the fastest dynamics that can be measured under analog acquisition. While the oscilloscope sampling rate can be chosen up to the GHz range, the DAQ analog inputs are often limited to a few MHz. Modern oscilloscopes can be

interfaced on a computer for direct collection of the data temporally stored in the oscilloscope (see Sec. IV A). In case of pulsed acquisition (see Sec. II C) averaging over several repetition is needed to obtain sufficient signal to noise ratio and/or to reduce the communication bandwidth requirement with the computer. Consequently, it demands a sync pulse for each repetition and limits the averaging strategy to the *NP* method described in Sec. III D.

Digital photocounters (d-PCs) deliver nanosecond pulses (*e.g.* TTL) with sharp rising edge. In such case, the use of a computer becomes almost unavoidable. Conventionally, a time tagging instrument or counter (*e.g.* from Swabian instruments or PicoQuant) provides time resolution down to picosecond range, below the typical timing jitter of the detector. Such resolution can be useful to resolve the PL lifetime (see Sec. V), or to acquire second order autocorrelation function ($g^{(2)}$) (*e.g.* to evidence single photon operations[98–100]). Otherwise, ODMR rarely involves that fast processes. Such a time precision can then become a disadvantage as the instrument buffer becomes more rapidly saturated by the recorded time tags. A field-programmable gate array (FPGA) can be used as a cost effective alternative to time-tagging instrument, but it requires dedicated programming skills. DAQ cards also include general purpose timers/counters. As presented in Sec. III A it is possible to configure them for photon counting with sampling rate up to twice their internal clock, which is two orders of magnitude faster than their analog inputs.

Finally, in the case of a wide-field configuration, the camera conventionally embeds its own ADC and directly delivers a digital image to the computer. In such cases, however, acquisition rates are at best up to hundreds of hertz, often too slow to resolve the pulse dynamics. When pulses cannot be isolated from each others, the averaging possibility is limited to the *NP* method described in Sec. III D.

### 5. Scanning in a confocal microscope

In a confocal microscopy configuration, locating individual emitters or mapping an area requires scanning the focal spot over the sample. Two possibilities arise:

- The sample or the microscope objective is mounted on an *xy* (or *xyz*) piezo-based nanopositioning stage so as to scan their respective position. In such case, the range is often limited by the stage itself from a few tens to a couple of hundreds of micrometers. If moving the objective, the field of view can also be limited by the shift of its back aperture with respect to the excitation and collection beams.

- The optical beam is angle-scanned by means of a 2-axis oscillating mirror and a $4f$ relay lens system. In such case, the field of view can be chosen by the $4f$ lenses and is ultimately limited by the field number of the microscope objectives.

While the first method is significantly less bulky, the second option is typically more affordable and offers a faster scanning rate. It can also offer the flexibility of changing the objective to adjust the field of view and resolution. Note that stepper motors and slip-stick piezo stages have bad repeatably: for accurate mapping and positioning, closed-loop positioners should be privileged. We also warn that magnetic materials should be avoided, in particular in the sample stage. In both cases, controlling the scanning device requires a computer program and a Digital to Analog Converter interface. Such functions are often provided via the aforementioned DAQ cards.

Apart from the essential components described in the previous paragraphs, the setup can be improved to allow additional characterization features. For example, sending the luminescence to a spectrometer allows to verify its origin and to quantify the amount of background emission (for example, for NV centers, emission from $NV^0$ is detrimental to the contrast). Moreover, to confirm that a single emitter contributes to the signal, a Hanbury-Brown and Twiss (HBT) interferometer can be used to measure the second-order autocorrelation function $(g^{(2)})$.

### B. Microwave instrumentation

The manipulation of electron spins is most readily achieved with near-resonant, coherent MW radiation. According to the targeted spin resonance(s) the MW source should have an appropriate frequency range (See Table I). Also, the phase (or frequency) noise of the source can be the limiting factor when studying highly coherent spin transitions. While a variable crystal oscillator (VCO) can be used as a very cost effective solution, tabletop MW generators such as from Rohde & Shwartz or Anritsu are often preferred in metrology and high-resolution spectroscopy for their spectral purity and power stability. A key specification for CW-ODMR is the dwell time in frequency sweep, which affects the acquisition time. MW powers above 30 dBm that can be achieved with standard MW amplifiers. An insulator should be placed after the MW source and/or the amplifier to prevent damage from back reflections.

MW pulses can be carved out of a CW MW source using external switches. Some MW generators may also directly embed an internal switch, which typically offers better performance to measure coherence times less than 100 ns due to lower phase noise between pulses. Furthermore, the MW source must be capable of frequency modulation (FM) for lock-in measurements. Compatibility between the various MW components must be checked.

The MW radiation is delivered to the sample by MW antennas. Working with ensembles may require spatially homogeneous MW amplitude over several hundreds of $\mu m^2$. Several designs focusing on optimizing their field intensity and spatial homogeneity can be found in[101–105]. Particularly, a trade-off needs to be found between the power conversion required to reach a fast enough Rabi oscillation frequency, the area of field homogeneity, and the bandwidth of the antenna (important when bias magnetic fields are applied to split the resonances). When spatial homogeneity is not essential, *e.g.* when working with a single spin, a small wire loop or straight wire can serve as MW antenna[106].

### C. Pulse generation

For time domain measurements (such the ones described in Sec. I D) a pulse generator must be included in the setup. While a minimum of two output channels is required to control both the MW and the laser, more channels can be useful for synchronization or for more versatile pulse schemes (such as rotating the spin around different axes on the Bloch sphere, or driving with multiple MW frequencies at once). The pulse generator must provide a minimum pulse length significantly shorter than the minimum coherence time to be measured. To allow multi-pulse protocols (*e.g* for Ramsey fringes), it must be possible to fire more than one pulse in each channel after a single trigger.

Two different options are available to produce optical pulses, starting with acousto-optic modulators (AOM). For a wide-field illumination, special care should be taken about the maximum optical admissible beam intensity to prevent any damage of the device. This is to be considered together with the desired rise and fall times, since all those parameters depend on the beam diameter on the AOM. As a handy alternative, laser diodes may embed pulsing capabilities. Unfortunately, overshoot can be present at the beginning of the pulse and depend on the preceding off state duration (through thermal relaxation). It may cause measurement artifacts, in particular when the repetition rate of the laser is varied, like in $T_1$ measurement (see Sec. I D). While post treatment mitigation methods are presented in Sec. V C, the use of an AOM reduces such artifacts.

| Component | Key specs | Ex. of Brand (and model) | Price (€) | Sec. |
|---|---|---|---|---|
| **Light source** | wavelength<br>intensity stability<br>transverse coherence<br>power (for wide-field)<br>intensity modulation option | any diode laser<br>Thorlabs (CPS 532)<br>Coherent (OBIS 532 nm)<br>Labs-electronic (DLnSec 520nm)<br>*Cobolt (06-MLD 515 nm)* | ∼ 300-7k | II A 3 |
| **AOM + RF driver** | diffraction efficiency<br>switching time (rise/fall) | Gooch & housego (AOMO 3350-199)<br>*AA Opto Electronic (MT350-A0.12-xx)* | ∼ 4k | II C |
| **Microscope Objective** | numerical aperture<br>working distance<br>Immersion medium (air, oil...) | Nikon, Leica, Zeiss, etc.<br>*Mitutoyo (0.6NA 1.3mm WD)*<br>*Olympus (LMPLFLN 50X 0.5NA)* | ∼ 500-5k | II A 2 |
| **Dichroic mirror / fluorescence filter** | cut-on wavelength<br>absorption losses<br>extinction outside transm. band | depending on the ODMR system<br>Semroc, Thorlabs, Chroma. etc. | ∼ 100-500 | II A 2 |
| **Light detector**<br>**Analog (a-PD)** | wavelength range<br>bandwidth<br>noise equivalent power | Newport<br>*Thorlabs (APD410A/M)* | ∼ 1k-2k | II A 1 |
| **Digital (d-PC)** | quantum efficiency<br>dark count rate<br>timing jitter | IDquantique (ID100), (ID120)<br>*Excelitas (SPCM-AQRH)* | ∼ 2k-8k | |
| **Camera** | pixel size/number<br>repetition rate, Gating<br>quantum Efficiency, readout noise | Thorlabs, UEye<br>Andor, Princeton instruments | ∼ 500-1500<br>∼ 2k-20k | |
| **Piezo stage** or **scanning mirror** | scanning speed<br>resolution, repeatability<br>travel range<br>closed vs. open loop | Thorlabs (GVS212/M)<br>*MadCityLab (Nano-3D200)*<br>*Newport (FMS-300)* | | II A 5 |
| **MW source** | frequency range<br>phase noise<br>sweep rate<br>power<br>pulse modulation | Wainvam (Wainvam-e1)<br>DS Instruments (SG4400L,SG6000L)<br>Anritsu (MG3691c), Agilent (n9310a)<br>Keysight (M9384B)<br>*Rohde&Schwarz (SMB100B)* | ∼ 500-600<br>50k | II B |
| **MW amplifier** | frequency range<br>gain and maximal power | *MiniCircuit (ZHL-16W-43-S+)* | ∼ 5k | II B |
| **MW antenna** | field intensity<br>bandwidth<br>spatial homogeneity | Ω-shaped antenna[101]<br>wire loop, straight wire<br>*Sasaki et al.*[102] | < 10 | II B |
| **MW switch** | insertion loss<br>switching time (rise/fall) | Minicircuit ZFSW-2-4,<br>Minicircuit ZASWA-2-50DRA+, | | II B |
| **Acquisition card (DAQ)** | maximum ADC sampling rate<br>number of i/o channels | Keysight-U2300A<br>*National Instrument (PCIe-6363)* | ∼ 3k-7k | II A 4<br>II A 5 |
| **Pulse generator** | minimum pulse width<br>number of channels<br>max. number of pulses | Spincore (PulseBlasterESR-PRO)<br>Zurich Instrument<br>*Swabian Instrument (Pulse streamer 8/2)* | ∼ 4k | II C |
| **Time-tagger** | minimum bin width<br>dead time<br>timing jitter | PicoQuant (PicoHarp300),<br>IDquantique (ID900) | ∼ 3k-10k | II A 4 |
| **Diamond sample** | bulk or nanodiamonds (ND)<br>NV density<br>NV coherence/relaxation time | *Element6*, *Appsilon B.V.* (bulk)<br>Van Moppes, Pureon,<br>Adamas nanotechnologies (ND) | ∼ 100-3k | |

TABLE II. Necessary equipment for a confocal ODMR instrument, with remarks and examples of suitable providers. The cells in gray color highlight specific equipment for pulsed experiments. In italic is what we used in our own NV setups. We underline the minimum equipment required to perform a ODMR confocal microscopy experiment.

## D. Bias magnetic field

Several applications require a bias d.c. magnetic field and precise controls of its strength and orientation, which allows to tailor the system eigenstates so that they are most susceptible to the perturbation of interest[107,108]. For example, when NV centers are used for magnetometry and temperature sensing, a bias axial magnetic field often allows to improve the sensitivity[44,109,110]. In addition, applying a field of the proper magnitude parallel to the quantization axis allows to work in specific level configurations such as at the ground state or exited state-level anti-crossing (GSLAC or ESLAC)[111]. Cross-relaxation conditions between different spin systems can be matched by properly tuning the magnitude of an axial magnetic field[112,113]. These configurations can be particularly relevant in the context of all optical sensing[112,114] or spin polarization[115]. Finally, a transverse magnetic field can be relevant in specific situations, *e.g.* for electrometry[107].

Different options are available and reported in literature to achieve this:

- Place a permanent magnet on a micrometer stage[109,116,117]. Controlling the direction, uniformity and intensity of the field can be challenging. Using a pair of permanent magnets mounted on a rotation stage can be helpful. Fine calibration is required for precise magnetic field alignment and repeatability.

- Design a system of three Helmholtz coils, each addressing one axis[118,119]. This approach requires more initial efforts, but leads to higher precision and control. It allows to generate a bias field in any possible direction and to swiftly switch between different bias fields. Ref.[120] presents a useful numerical model to simulate the magnetic field in the center of the coils for varying parameters. If the bias field needs to be modified during the measurement (as for example in real time electrometry[43,121]), this approach is particularly convenient.

- Hybrid solutions, matching the specific application requirements[114,121–123]. It is often not necessary to generate a bias field in any possible directions. Simpler systems compound of one or a pair of Helmholtz coils and permanent magnets are often sufficient. Placing the sample on a rotation stage can offer more convenience and versatility.

To conclude this Section, we refer to Table II that summarizes hardware requirement for a typical room-temperature confocal ODMR setup.

## III. CONNECTION AND SYNCHRONIZATION

We previously mentioned that the visualization of ODMR signals can be either performed on an oscilloscope or through an acquisition card on a computer and we described when the second option is necessary. Here we explain how to connect and synchronize all devices in either case, focusing on four typical ODMR measurements.

## A. Photon detection

As mentioned in sec. II A 4, two main types of detectors (analog photodectors, a-PD vs. digital photon counters, d-PC) can be employed depending mainly on the signal intensity. The required connections for each case are illustrated in Fig. 6. On the one hand, the analog signal coming from an a-PD can be measured either directly with the help of an oscilloscope or on a computer after digitization via DAQ. In the latter case, the a-PD is plugged to an analog input. The analog acquisition is limited to the clock frequency of the ADC, $f_{ADC}$, such that the temporal resolution cannot be better than $T_{Sampling} = 1/f_{ADC}$. For example, in the case of the NI-6364 card and for single channel, $T_{Sampling} = 0.5\,\mu s$, increased to $n_c \times T_{Sampling}$ when $n_c$ channels are used.

On the other hand, the output of a d-PC is a train of pulses (*e.g.* TTL), each corresponding to one (or more) detected photon. A DAQ can be used to measure this signal. Commonly, one of its internal counters is configured to count the number of received pulses through one of its PFI (Programmable Function Input). The count is then periodically stored in the DAQ buffer and reset. As a result, the number of counts per interval is obtained. In such a case, the sampling rate can be similar to analog signals.

In order to increase the temporal resolution, which is most desired in pulsed measurements, three counters of DAQ can be used for time tagging. Two counters are set at a high data rate as inputs to count the arriving signal from the d-PC and sync signal that triggers the acquisition. The third counter is set as output and it generates a periodic TTL signal to internally reset the other two. Their count are then saved in the buffer after receiving the TTL reset signal. The reset signal determines the temporal time-tagging resolution. Knowing the relative time difference between sync and signal, the averaged temporal signal can be reconstructed.

Alternatively, the role of the clock and the d-PC pulses can be inverted. The pulse count is then increased by one at each period of the clock, while each pulse, when detected, triggers its storage to the buffer. This results in a list of time-stamped events. At the cost of more information saved, which can slow down the computer, this architecture can achieve a better time resolution. With the same NI-6363 card considered above, one can achieve the temporal resolution of $T_{Sampling} = 1/(2 * f_{counter_{clock}}) = 5\,ns$; the factor 2 accounts for both rising and falling edges being counted.

## B. Confocal scanning

In confocal microscopy, a scanning device (such as a piezo stage or a scanning mirror) is controlled while the sample PL is monitored. Synchronizing the two allows to attribute to each pixels its corresponding PL intensity, creating an image. As illustrated in Fig. 7, these controls and PL monitoring

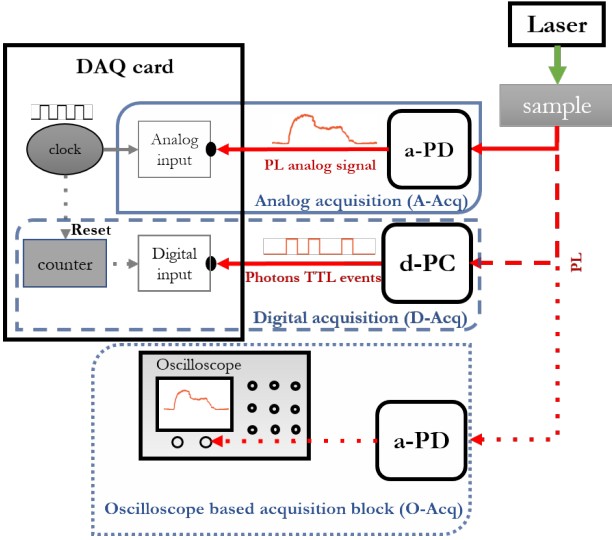

FIG. 6. Hardware connections for monitoring a photoluminescence signal. The solid and dashed/dotted lines indicate the three hardware alternatives. The green color refers to the pumping beam and the red to the sample photoluminescence (PL). DAQ card internal connections are shown in gray. The analog, digital and oscilloscope-based acquisition blocks introduced here, are also used in Figs. 7, 8, and 9 along with the same conventions.

are commonly performed with a DAQ card in which an analog output is used for each spatial axes $(x), (y)$ and $(z)$. The measurement is performed by generating a sync signal of frequency $f_{sync}$ in the acquisition card and three step-ramp functions ranging from the lowest to the highest voltage are sent to the control ports of the scanning device. Each sync period therefore corresponds to a specific position of the scanning device for which the PL is collected. The sync frequency and order of the generated step ramp functions are eventually used to build the PL image.

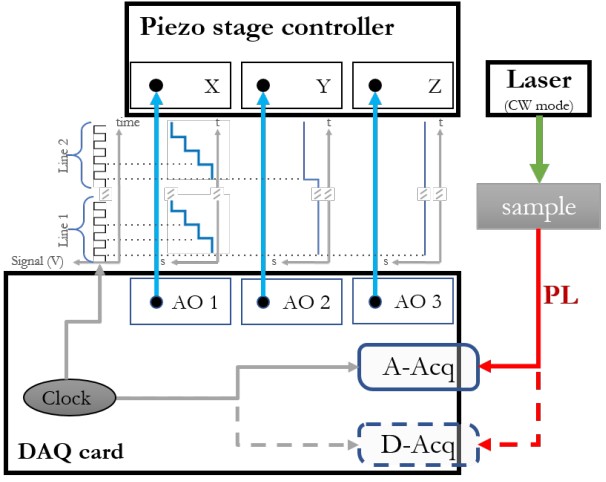

FIG. 7. Hardware connections and synchronization links for confocal scanning. The analog and digital data acquisition blocks are introduced in Fig. 6.

## C. CW ODMR

Continuous wave ODMR spectra are measured in a similar way, with the difference that the sweep is applied on the MW frequency instead of the beam (or sample) position. Each optical intensity must be attributed to the corresponding MW frequency. As described in Sec. I D, the plot of the PL versus MW frequency mirrors the magnetic resonance spectrum. As reported in Fig. 8 the DAQ sends a TTL sync signal to the MW source and triggers frequency change while the corresponding PL signal is recorded simultaneously.

In case of using an oscilloscope for acquisition, the MW source, set in frequency modulation mode, becomes the synchronization master. It sends either a sweep or a TTL signal with the same period to a second input of the oscilloscope to trigger at the beginning of each sweep. The horizontal temporal scale of the oscilloscope can then be adjusted to display the PL as a function of the frequency. In the case of using a VCO, the frequency has to be set by an external voltage source, such as generated by a DAQ or waveform generator. The visualization on an oscilloscope can be performed in the same way.

Typical MW sources offer two modes to change the MW frequency: sweep and list. In the sweep mode, the minimum and maximum frequency and the step size are defined. As the MW generator receives a TTL signal, it changes the frequency by the defined step. In the list mode, the list of frequencies are previously stored in the generator, then as a TTL signal is received, the frequency is changed to the next one in the list. The list mode allows to perform random sweeping of the MW frequency mitigating problems such as hysteresis due to frequency-dependent MW heating through the antenna resonance. In practice, the acquisition speed of a CW-ODMR spectrum is limited by how fast the MW generation hardware can switch the output frequency. This values is for example around 1 ms minimum per step for R&S SMF100A.

## D. Pulsed measurements

As depicted in Fig. 9, pulsed measurements require synchronizing the laser and MW pulses with the optical signal acquisition. A pulse generator sends a TTL signal to command the laser and MW output switch. An additional TTL signal is also generated within the pulse generator to trigger the acquisition and set the time reference for each measurement. This signal is sent to the DAQ or oscilloscope. For each data point of a pulsed experiment, the optical signal is analyzed against the change of one variable (*e.g.*, duration of the MW pulse or wait time between MW and laser pulses, see Sec. I D).

*Averaging strategies* — We discuss two different options, where $i \in \{1; P\}$ labels the value of a parameter (*e.g.* waiting time) to be swept among $P$ values and $N$ is the number of repetition over which the signal is to be averaged.

- A first averaging method is denoted as "*NP*", where each $i^{th}$ pulse sequence is individually repeated $N$ times,

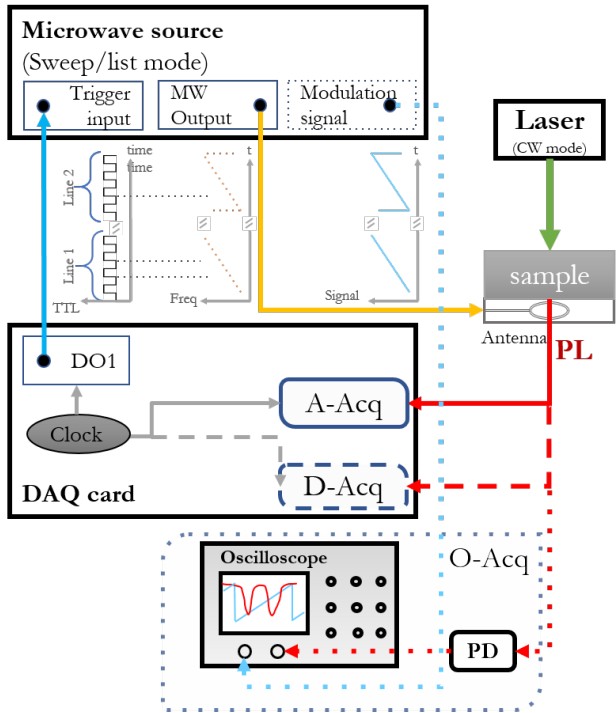

FIG. 8. Hardware connections and synchronization links for CW-ODMR, MW connections are shown in orange. Other conventions are the same as in Fig. 6.

before moving to the next parameter value from the $1^{st}$ to the $P^{th}$.

- In the second one, denoted as "*PN*", a full series of $P$ sequences (consisting of $P + 1$ optical pulses since the first only serves as initial polarization) is sent at once while the PL is recorded. This sequence is then repeated $N$ times to reach the desired signal to noise ratio. In this method, a slow varying noise similarly affects signals corresponding to all parameter values and is therefore averaged out. It also allows to monitor how the signal to noise ratio improves over the successive repetitions. However, it demands sufficient time resolution to isolate signals from each pulse sequence making it hardly compatible with the use of a camera. It also requires digitizing and storing the full sequence at once, which also limits the use of an oscilloscope (see Sec. II A 4).

*Synchronization strategies* — In either cases, two distinct synchronization methods are possible:

- 'Sync method 1': Only two TTL sync pulses are generated, at the start and at end of each series of $N$ (or $P$) sequences. The incoming optical signal is continuously recorded between the initial and final pulses according to the sampling rate. The position of each pulse is then calculated by software (see Sec. V C). This cannot be done when using an oscilloscope.

- 'Sync method 2': A TTL pulse is sent at the beginning of each pulse to be either timestamped by the counter

with a DAQ card or used to trigger the oscilloscope.

Fig. 9b depicts two out of the four possible combinations of averaging and synchronization strategies in the case of a Rabi oscillation measurements.

*Implementation with a DAQ card* — The implementation of aforementioned methods with an NI card can be elaborated based on the type of the acquired signal.

- In the analog case, the a-PD signal is sent to an analog input channel of the DAQ card. As discussed in section III A, the ADC captures and saves the analog signal into the buffer until it is read out. With 'Sync method 1', the generated sync pulses are transmitted to another analog channel of the DAQ card and the acquisition starts as soon as it receives a sync pulse. As a consequence, the optical signal is recorded even when the laser is off. Identification of each pulse then requires post-processing as described in Sec. V. With 'Sync method 2', the generated sync pulses are sent to a digital channel of the DAQ card. The channel is adjusted to acquire the pulses during the laser pulse windows using start edge trigger and pause trigger protocols of the DAQ card.

- Acquiring data from a d-PC, however, demands time-tagging as described in section III A. Similarly, the 'sync method 1' records all the photon received. the pulses are found and reconstructed by post-processing. With 'sync method 2' the trigger signals are time-tagged and mark the beginning of each pulse.

One chooses the synchronization method depending on hardware limitations and desired averaging strategy discussed above.

## IV. QUDI, AN OPEN-SOURCE INTERFACE FOR ODMR EXPERIMENTS

As discussed in Secs. II and III, the use of a dedicated software can be necessary to control a confocal microscope or deal with photon counting. It can also automatize the control of ODMR instruments to speedup the measurements and ease their reproducibility, either for fundamental studies or application developments. Ideally, the software interface should deal with both single photon counters in low light conditions and analog photodetectors, process confocal and wide-field images, and freely combine different measurement parameters and methods.

Several options exist for coding such interface program. On the one hand, mid-level languages such as C++ or FOR-TRAN, down to the lower-level ones such as assembly, offer the best possibilities to optimize computer resources. They are often the best option for demanding computational physics and simulations, artificial intelligence related research, or for processing of complex or voluminous data[124]. On the other hand, higher level programming languages, such as Python

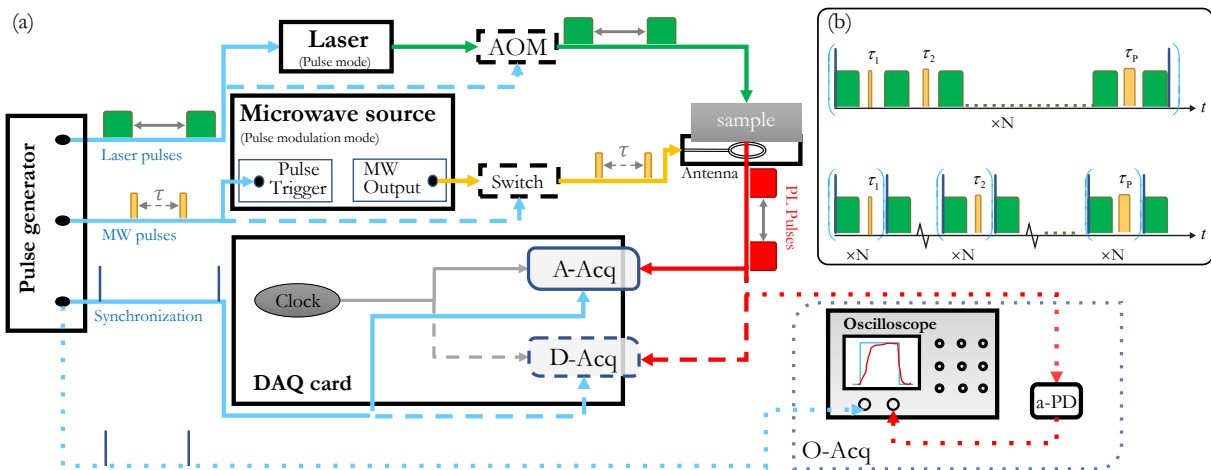

FIG. 9. (a) Hardware connections and synchronization links for pulsed experiments. Solid and dashed lines indicate different alternatives depending on available hardware. (For example, if the laser permits pulsed modulation, the AOM is not required.) (b) Averaging and synchronization strategies in the case of a Rabi measurement. Top: *PN* strategy with 'Sync method 1', Bottom: *NP* method with 'Sync method 2'

or Java, greatly facilitate design, development and maintenance of codes. They also ensure better portability between different hardware configurations. They are therefore particularly appropriate for interfacing laboratory experiments[125,126]. A comparison between programming languages including C, C++, Java and Python can be found in[127].

Among others, Labview (for Laboratory Virtual Instrument Engineering Workbench), a proprietary programming environment developed by National Instruments[128], offers a graphical diagram coding style making it more intuitive to newcomers in programming. Moreover, it permits to easily interface with laboratory instruments and offers libraries for signal generation, data acquisition and processing that allow to rapidly setup basic experiments. However, when the experimental complexity increases, this graphical programming style can become difficult to handle. It also remains incompatible with versioning and collaborative tools such as Github for sharing codes and subroutine within a community.

In this context, Python, a high-level programming language, offers a great alternative[129]. Coding environments and interpreters being open source, it is particularly popular and used in various domains, including experiment control. It supports versatile and efficient community-developed libraries. Moreover, it offers sufficient level of abstraction to allow portability between computers of different configurations and making it upgradeable to new functionalities, embeds efficient multi-threading and signaling capabilities allowing for parallel operations (data acquisition, processing and display).

Here, we introduce *Qudi*, an open-source Python code dedicated to ODMR measurements. First presented in[19], it is an open-source collaborative project shared on Github and it benefits from a well programmed Python-based architecture. We first present Qudi's original features and the ones we added to improve its performance and extend its applicability. We then detail *Qudi*'s architecture and presents how to configure and

use it. The official *Qudi*'s version is available on Github[130] and a general documentation based on developers "in code" comments along with a brief general tutorial can be found here[131]. For those not already merged within the official Github, the added features can be found on our Github page forked from it[132].

## A. Qudi's features

Qudi[19,130,132], initially released in 2017, integrates modules for performing ODMR experiments, instrument control, and real-time data acquisition and processing. It enables fast imaging by scanning the beam position while collecting luminescence. Qudi can locate ODMR emitters and track them during acquisition, compensating for sample drift. It is particularly useful for studying nanoparticles in liquid suspensions[133,134]. It provides useful tools for interfacing a scanning confocal microscope by controlling piezo-based nanopositioning stages or scanning mirrors and collecting the optical emission signal. Possibilities include in-plane scanning (*xy*), *z*-stack acquisitions (to obtain 3D images), out of plane cross-sections (*xz*) or (*yz*) as well as any other direction cuts depending on the wiring and instrument. Associated to confocal microscope hardware (see Sec. II A), Qudi can locate and track ODMR emitters, compensating for sample drift. This feature is particularly interesting when dealing with nanoparticles in a liquid suspension such as in biomedical applications[133,134].

Qudi efficiently controls the instruments required for acquiring CW ODMR spectra and determining spin resonance frequencies. Real-time fitting of analytical curves on ODMR data allows for adjustment of acquisition parameters and quick sharing of results with collaborators. In addition to the CW ODMR signal, Qudi is designed for the study of spin dy-

namics with pulse sequences such as for measuring longitudinal spin relaxation curves ($T_1$), Rabi oscilations, Ramsey ($T_2^*$) or Hahn Echo ($T_2$) sequences (see Sec. I D). It can control the pulse generator for triggering laser and MW pulses (see Sec. II C) in a synchronous manner with the collection of the photoluminescence (see Sec. III D). It also enables the design of complex pulse sequences, including dynamical decoupling sequences. Predefined methods like XY8 pulses are available via the graphical interface, and new pulse sequences can be easily added to the program[135].

Qudi offers tools for on-the-go pulse extraction, analysis, and automated experimental parameter settings. This simplifies long measurement sequences across different materials. Using Qudi's Jupiter notebooks[136], one can design and automate independent experiments to vary multiple control variables, enabling efficient exploration of parameter space.

The initial version of Qudi was focused for experiments on single quantum emitters, thus requiring digital single photon counting modules as well as time tagging electronics in the pulsed measurements. In order to broaden its contextual applicability, we added several features.

At first, a direct oscilloscope interface was implemented for data acquisition (see Sec. II A 4) while, conversely, NI card (or similar DAQ device)'s digital inputs can now also be used as a fast event counter and the minimum temporal resolution has been reduced to half the card internal clock period, making it sufficient for ODMR experiments without need of dedicated time tagging instruments.

Two strategies for photon counting were implemented (see Sec. III A) and a new logic for pulsed measurements was coded to enhance acquisition speed and eliminate the need for post-treatment and pulse extraction described in Sec. III D. Pulsed measurements using analog signals were also added, making the system compatible with analog photodiode signals for studies of ensemble (see Sec. II A 1). Analog acquisition in CW-ODMR was also improved by increasing the sample rate per frequency sweep, optimizing signal-to-noise ratio (see Sec. III A).

Finally, an arbitrary/random MW frequency sweep was added to minimize thermal effects of the resonator on the ODMR signal and an optical spectrometer module was incorporated for ODMR signal acquisition using a user-specified spectral window. The summary of the added features to the Qudi platform is shown in Table III.

## B. Main architecture

Qudi is made of different modules that are loaded and connected together by a manager component. The science modules responsible for experiment control are divided into three categories: GUI (graphical user interface), logic, and hardware. As detailed in the following, this division is based on a clear separation of tasks among the categories and offers a reliable and flexible software architecture and simultaneous (multi-threading) acquisition, data treatment and visualization. Importantly, the setting of those modules and the links between them is to be defined in a configuration file (See Sec. IV C).The important notion of hardware interface (higher abstraction level of hardware module) is also introduced.

### 1. Modules

*GUI modules* — They create an interactive graphical interface that allows the user to control the experiment and to visualize the acquired data. In particular, the GUI allows to start and stop the acquisition, to adjust the experimental parameters (*e.g.* MW power, frequency sweep range, acquisition time...), to save the data and to fit them with pre-defined functions, simply pushing buttons or inserting values into the interactive windows. Changes made in the graphic interface, such as a pushed button or a modified parameter, trigger a broadcast signal sent to the related logic modules. Although GUI modules offer an intuitive and efficient way to interact with the logic modules, Qudi can also be fully functional without them via the integrated Ipython console or via a Jupyter notebook.

*Logic modules* — They form the basis of every experiment designed in Qudi. Their main function is to coordinate the different tasks necessary to perform a complete experiment. They serve as a bridge between the two other kinds of modules. On one side they receive the emitted signals from the GUI modules and they send others to the hardware modules to configure the instruments[137]. Conversely, they gather and analyze data coming from the hardware modules and pass them to the GUI modules for display.

Each experiment depends on a variety of generic tasks that are common to different types of measurements, such as fitting appropriate curves, saving data, etc. Therefore, rather than having a single logic module for each experiment performing all these tasks, they are divided in multiple logic modules used in a versatile manner. Besides communication with hardware and GUI modules, each logic module can receive inputs from and send outputs to other related logic modules. Note that logic modules are the only ones allowed to talk to modules of the same category.

*Hardware modules* — The main task of these modules is to allow an effective communication between the logic modules and the specific hardware. Receiving orders from the logic modules, they run the equipment drivers (often provided by the supplier) to act onto the experiment. They also send the output signals of the instruments to the logic modules, to be further analyzed, displayed and eventually saved. Moreover, hardware modules can also work as virtual dummy or mock hardware, emulating the functionality of a real device. This possibility can be very helpful for configuring a new setup. As an example, modules and links between them that are required for the four typical experiments described in Sec. III are presented in Figure 10.

### 2. Hardware interfaces

Even if a set of instruments have common features for performing similar tasks, there is, between different models and

| Module | Original features | Added features |
|--------|-------------------|----------------|
| **Counting** | Analog acquisition<br>Digital acquisition | Acquisition with oscilloscope<br>Faster analog and digital DAQ acquisition |
| **Confocal** | Analog acquisition<br>Digital acquisition | Acquisition with oscilloscope<br>Control with *Pulse Streamer* |
| **ODMR** | Analog acquisition<br>Digital acquisition | Arbitrary/random MW sweep<br>Acquisition with oscilloscope<br>Acquisition with Spectrometer<br>Lock-in detection* |
| **Pulsed** | Digital acquisition with time tagging device | Analog acquisition with DAQ<br>Analog acquisition with oscilloscope<br>Digital acquisition with DAQ<br>Pulse extraction improved<br>New pulse sequences |

TABLE III. Summary of Qudi main features in the original release[130] and in our upgraded version[132]. *Under development.

suppliers, a wide difference in command structure, grammar and connection methods. In order to handle this complexity, a specific interface is defined for each category of instruments. In the context of object oriented programming, hardware modules are classes that inherit from hardware interfaces. The running of a laboratory equipment consists in defining and operating instances (objects) from the instruments-related classes. The interface therefore consists of a set of functions that each hardware module within the same category must implement. Therefore, each instrument used in an experiment should have its own class compatible with the related interface class in Qudi. This class needs to control the instrument while respecting constraints on its operation as specified in its data sheet. To accomplish this task, the class may use drivers provided by the instrument's manufacturer. Various instruments have already been introduced in Qudi in different categories (such as spectrometers, cameras, MW generators, pulse generators, etc.). However, due to the wide variety of systems, applications, suppliers and models available, it may be necessary to add a new device or modify the class for an instrument whose model is slightly different from the one defined in the code.

### C. Utilization

Installing Qudi is rather straightforward from Qudi Documentation's instructions[138]. It can be cloned from the original[130] or our[132] Github.

Performing experiments with Qudi, also requires to tune it specifically for the user's hardware environment and specific needs by writing a configuration file, which, in particular, defines the connections between the modules that are used such as represented in Figure 10. Besides, installing an instrument that is not yet supported requires to write the corresponding hardware module class (see above section IV B 1). Finally, for important coding project such as Qudi, it can be useful to configure the Python environment to ensure the good functioning of the different packages and libraries that are used in

different places of the code. This can be performed through a `conda` environment installer such as the one called `Anaconda` available in the Github subfolder `qudi\tools`.

The utilization of the module we added in our Github[132] (see Sec. IV A) is detailed in[139],[140].

In the following section, we will focus on the characterization of an NV center ensemble to propose some practical advice and discuss good practices when performing typical ODMR experiments with a home-build confocal microscope (see part. II) controlled using Qudi.

### V. GOOD PRACTICES FOR ODMR MEASUREMENTS: NV CENTER CHARACTERIZATION

As described in Sec. I D the main figures of merit characterizing an ODMR system are the longitudinal spin relaxation time $T_1$, dephasing time $T_2^*$ and coherence time $T_2$ (the latter being dependent on the type of echo sequence used to measure it). In this section, we describe the procedure and methods to reliably perform these measurements such as presented in Fig. 3. As an illustration, we characterize a small ensemble of NV centers in bulk diamond using a single pixel detector in a confocal geometry (see Sec. II A 1). While an extensive review on optimizing sensitivity with NV centers can be found in[44], we aim here to provide practical advice on how to easily reach good measurement quality in terms of acquisition speed, contrast and repeatability.

The measurements presented in this section are performed on a bulk diamond sample enriched in NV centers and provided by Appsilon B.V. Other possibilities for commercial diamond providers are summarized in Table II. It should be noted that the figures of merit discussed here can vary significantly depending on the material quality. For practical purposes, we provide throughout the section typical values for these figures of merit in an NV ensemble. They are given for a sample at room temperature, in the absence of external perturbation (magnetic field, electric field, pressure, etc.).

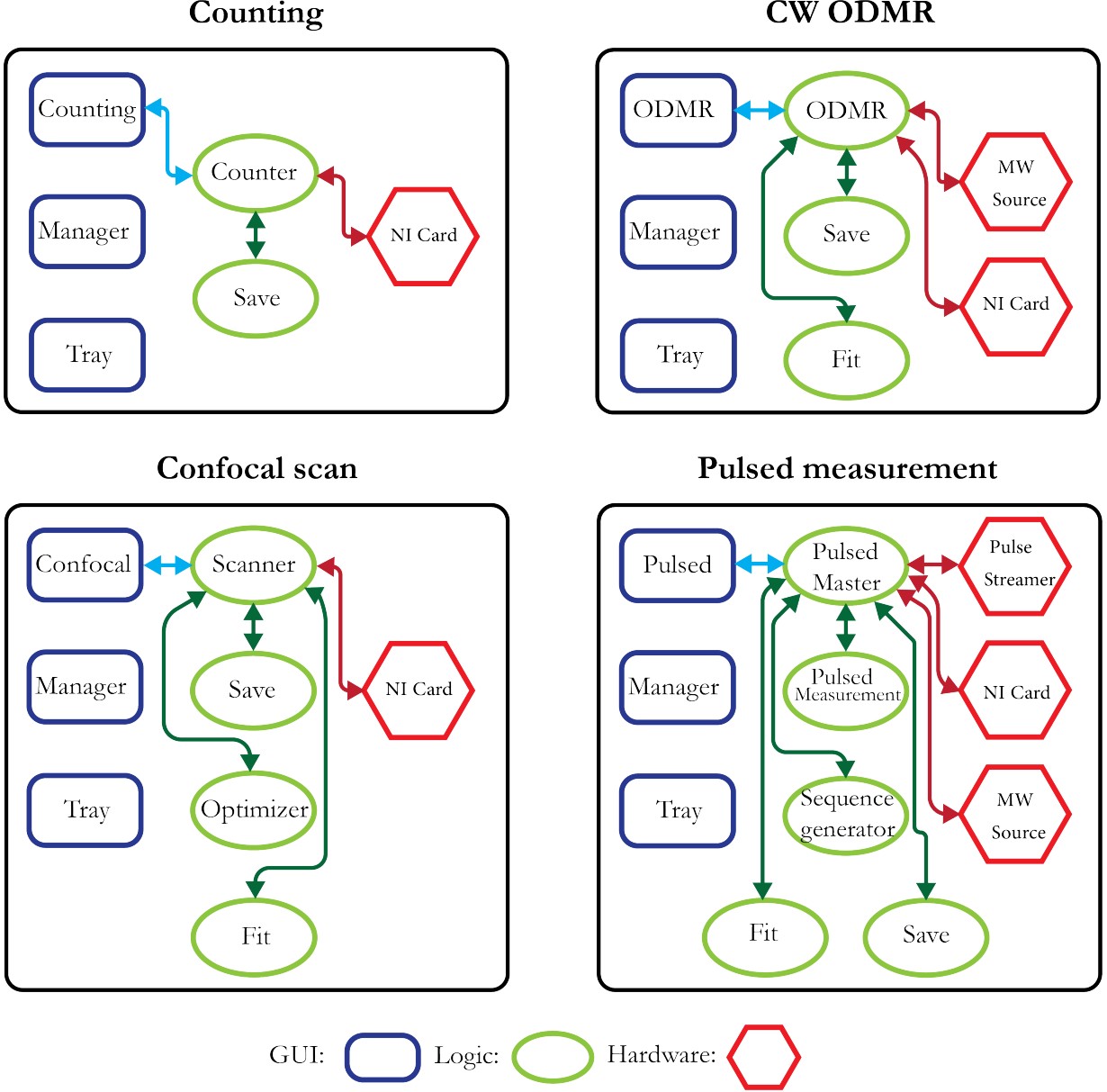

FIG. 10. Block diagrams for the software connections of counting, confocal scan, CW ODMR, and pulsed measurement which needs to be included in the configuration file.

## A. Procedure

The sample is first placed in the optical setup and its position adjusted to match the object focal plane of the objective. A confocal scan allows to image the sample and identify the region of interest. At this step, further alignment of the setup to ensure that maximum signal is emitted by and collected from the NV centers is possible. A single NV center can provide photon count rates above 100 kHz when using a numerical aperture exceeding 0.9 and a laser excitation power in the mW range. The total PL depends linearly on the number of NVs in the ensemble. According to the level of brightness of the sample and collection efficiency of the setup, a proper

detector (*e.g.* photon counter or analog photodiode) and possibly a neutral density filter are selected. Then, we use Qudi ODMR interface to perform a CW-ODMR scan. The goal of this step is to adjust the bias magnetic field (by monitoring the different resonances in the spectrum) and to identify the resonance frequency that will be used for the following pulsed measurements.

One specificity with NVs is that each center can take one of the four (111) crystallographic orientations of the diamond lattice. Considering the two allowed MW transitions from $|m_s = 0\rangle$ to $|m_s = -1\rangle$ and $|m_s = +1\rangle$, this can lead to eight distinct resonances (that are further split by hyperfine interaction) whose frequencies correspond to the projection of the magnetic field along each orientation. It allows to per-

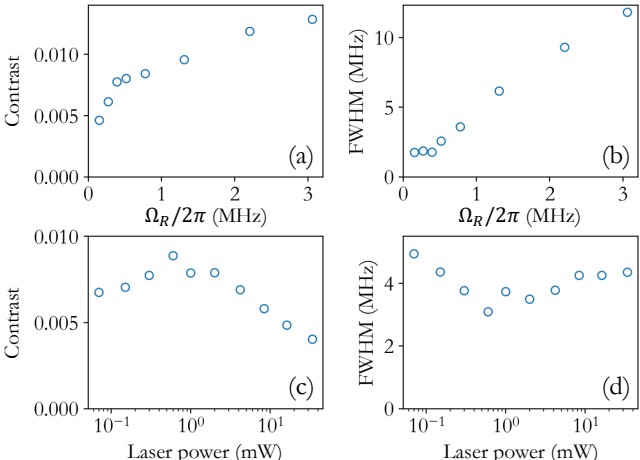

FIG. 11. (a, c) Contrast and (b, d) FWHM of a single resonance in the CW ODMR spectrum, versus (a, b) MW Rabi frequency and (c, d) laser power.

form vectorial magnetometry[85,141], even without biased applied fields[142,143]. Certain applications require a specific magnetic field alignment (see Sec. II D). In this example, we target a bias field such that the Zeeman splitting for at least one NV orientation is different from all other orientations, *i.e* at least one pair of transitions in the spectrum is non-degenerate.

We select a MW frequency resonant with one of these two transitions (corresponding to either $|m_s = -1\rangle$ or $|m_s = +1\rangle$) and perform pulsed measurements to characterize $T_1$, $T_2$ and $T_2^*$. To this end, we move to Qudi interface for pulsed Measurement. Rabi oscillations are first measured to determine the duration of a $\pi$-pulse which is half the oscillation period. We note in passing that the decay of Rabi oscillations is not simply related to any of the time constants mentioned above because increasing the MW pulse duration reduces is spectral bandwidth and thus progressively filters a smaller set of NV centers among the inhomogeneously broadened ensemble. Since a good estimate of $T_2^*$ can be obtained from the CW-ODMR linewidth (in the limit of low MW powers) it is recommended to use sufficient MW power in the pulse sequences so that the $\pi$-pulse is shorter than $T_2^*$ in order to efficiently drive all NV centers in the ensemble. The last step consists in applying each of the dedicated pulse sequences to extract $T_1$, $T_2$ and $T_2^*$, previously presented in Sec. I D.

## B.   CW ODMR

In order to perform CW ODMR with highest possible contrast and narrow linewidth, a compromise must be reached in terms of both MW and laser powers, as illustrated in Fig. 11. While the optical excitation rate must be faster than the $1/T_1$ relaxation rate to ensure that all spins are well polarized, a too high rate can decrease the ODMR contrast by competing with the MW excitation and preventing a large population difference between states $|m_s = 0\rangle$ and $|m_s = 1\rangle$[144]. On the other hand, while increasing the MW power first enhances

the contrast, it then broadens the resonances through power broadening[80] and eventually reaches saturation of the spin transition.

As practical guidelines (for NV ensembles) the optimal laser power is the one maximizing contrast and the optimal MW power is the highest power for which the linewidth remain close to the $T_2^*$ limit. For further optimization, the optimum condition for the laser and MW power can be derived analytically using a rate equation approach[80,144]. Typically, a single NV center exhibit an ODMR contrast above 10%. For an ensemble with 4 possible NV orientations, this results in best contrast of around 2% for each resonance in the ODMR spectrum, close to the values reported in Fig. 11. The presence of other photoluminsecent defects, such as neutral NV0, causes a reduction of the ODMR contrast.

## C.   Pulsed experiments

To improve the quality of a pulsed experiment several factors should be taken into account. Some are general to all pulsed protocols, while others are more specific. Common requirements for all pulsed experiments are reliable initialization and readout. Laser pulse duration needs to be long enough to ensure maximum polarization in state $|m_s = 0\rangle$. The characteristic time for spin polarization depends on the optical power density and typically ranges from 100 ns to 1 μs. For NV centers, the initialization pulse must be followed by a waiting time, typically 1 μs, to allow relaxation of population trapped in the singlet state towards the ground state $|m_s = 0\rangle$[145]. The characteristic polarization time also impacts the readout phase. Depending on the time resolution of the hardware (see Sec. II A 4 and III A), it can be convenient to reduce the laser power, to polarize the spin slower and be able to resolve it dynamics as in Fig. 12. The duration of the optical pulse needs to be adjusted accordingly.

In the case of 'Synchronization method 1' (see Sec. III D) in which only the beginning and the end of a sequence is triggered, it is necessary to locate each pulse within the entire recorded optical signal. In this case, two standard approaches are implemented in Qudi for pulse recognition. The first uses a threshold value to identify rising and falling edges of a pulse. The second relies on a convolution of the signal with a Gaussian distribution and takes its first derivative, resulting in a pulse identification signal $S_p(t)$ given by:

$$S_p(t) = \frac{d}{dt}\left[ S_r(t) \otimes \frac{1}{\sigma\sqrt{2\pi}}e^{-\frac{1}{2}\left(\frac{t}{\sigma}\right)^2} \right] \tag{8}$$

where $t$ represents the time and $S_r(t)$ is the raw signal acquired during the pulsed measurement; $\sigma$ is the standard deviation for the Gaussian filter. Rising and falling edges of pulses correspond to local maxima and minima of $S_p(t)$. The Gaussian filter allows to smooth the signal and reduces false pulse detection. This second method is thus generally more robust than the first one. Optimal extraction is achieved by adjusting the parameter $\sigma$.

Figure 12 displays the time evolution of luminescence intensity during a laser pulse starting at $t = 0$, for spins initially

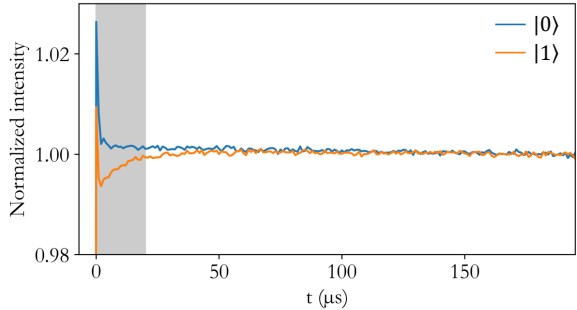

FIG. 12. Normalized luminescence intensity during the readout laser pulse, for spins initialized in state $|0\rangle$ (blue curve) and $|1\rangle$ (orange, using a $\pi$-pulse between optical pumping and readout), collected with NA 0.65 objective and 35 mW incident laser power. The shaded gray region indicates a good choice of time window for signal integration, yielding high SNR.

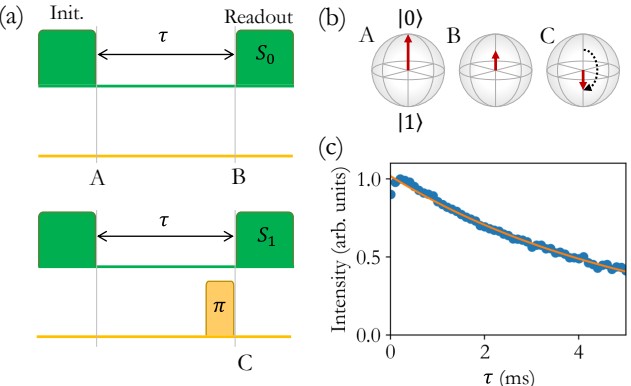

FIG. 13. (a) Pulse protocol for optimized $T_1$ measurement, with the addition of a $\pi$-pulse to extract $S_1$. (b) Bloch sphere representation of the spin state at times of the pulse protocol indicated in (a). (c) Resulting signal versus delay $\tau$, fitted with an exponential decay (solid line)

prepared either in $|m_s = 0\rangle$ or $|m_s = 1\rangle$. The different levels of PL in the two cases and the spin polarization dynamics, are clearly resolved. To obtain a measure of the spin projection along the quantization axis the PL intensity is integrated over an early time window marked as gray shaded area, and normalized to the PL intensity integrated at the end of the pulse (not represented), when spins are polarized again. It allows to compensate possible slow fluctuations of the laser intensity or collection efficiency.

To maximize the signal to noise ratio, the time window should start with the laser pulse, where the PL difference between $|m_s = 0\rangle$ and $|m_s = 1\rangle$ is maximal. If the initial time of a pulse is not properly identified by the pulse extraction method for all pulses in the sequence (*e.g.* due to slow laser rise time), the resulting timing jitter translates into noise on the total integrated signal. In this case excluding very early times of the readout pulse can be beneficial.

Several other sources of noise can affect readout fidelity. Among them, some are spin-independent, for example due

to NV ionization and NV- recombination caused by the laser pulse[146,147]. A powerful solution to suppress this kind of noise is to repeat the pulsed protocol but inverse the spin state onto $|m_s = 1\rangle$ just before readout using a $\pi$-pulse as illustrated in Fig. 13 [148]. The extracted signals with and without inserted $\pi$-pulse are labeled $S_1$ and $S_0$, respectively. The expectation value of the spin state projection is linearly related to the difference signal $S_0 - S_1$ from which spin-independent common noise is removed. The result of this procedure for a $T_1$ measurement is illustrated in Fig. 13.

Another important source of artifacts arises when the dark time $\Delta T$ between consecutive laser pulses is swept. As mentioned in Sec. II C it can lead to variation in laser pulse intensities over for the entire sequence. In most cases, a simple solution is to keep $\Delta T$ constant, adjusting some waiting times before and after the microwave pulses accordingly, as long as $\Delta T \ll T_1$. It is however not possible for a $T_1$ measurement. Fortunately, the protocol discussed just above allows to partially mitigate such artifact by normalizing the signal by $S_0 + S_1$. Such protocol also allows to extract a relevant signal even when the setup temporal resolution does not allow to resolve the pulse dynamic, *e.g.* when using a camera in a widefield configuration.

A final knob for optimization of pulsed measurements is in the definition of $\pi$, $\pi/2$ and other pulses. The duration of such pulses is in principle extracted from the measured period of Rabi oscillations. However, in practice small deviations can arise, *e.g.* due to not perfectly square MW pulses and fitting errors. It is thus good practice to perform a calibration step. To this end, we use a Hahn echo sequence with fixed delay and monitor the echo amplitude for varying initial, central and final MW pulses. The result of this procedure is illustrated in Fig. 14, where the initial and final pulses are half the central pulse. Every second sequence, we also replace the final pulse with a $3\pi/2$-pulse, to apply the above common noise rejection strategy. The optimum pulse duration corresponds to the simultaneous maximum echo amplitude when the final pulse is $\pi/2$ and minimum when it is $3\pi/2$. This ensures maximum contrast for a $T_2$ measurement, using the Hahn echo sequence (Fig. 3(d)). The calibrated $\pi/2$-pulse is also used for $T_2^*$ measurement using Ramsey interferometry (Fig. 3(c)).

As stated earlier, $T_1$, $T_2$ and $T_2^*$ depend greatly on the diamond quality, as well as on the presence of external perturbations. For an NV ensemble with moderate concentration (below 1ppm), $T_1$ is usually on the order of several milliseconds, $T_2^* \approx 0.5 - 1\mu s$ and $T_2$ is limited to few 10s of $\mu s$ with a simple spin-echo sequence, but can be extended to several 100 $\mu s$ with better dynamical decoupling protocols. The interested reader should refer to e.g. Ref.[44] for a more quantitative, in-depth discussion of these coherence times, which is beyond the scope of the present paper.

## CONCLUSIONS AND PERSPECTIVES

In this article, we present the settings of an experimental breadboard for ODMR experiments with both commer-

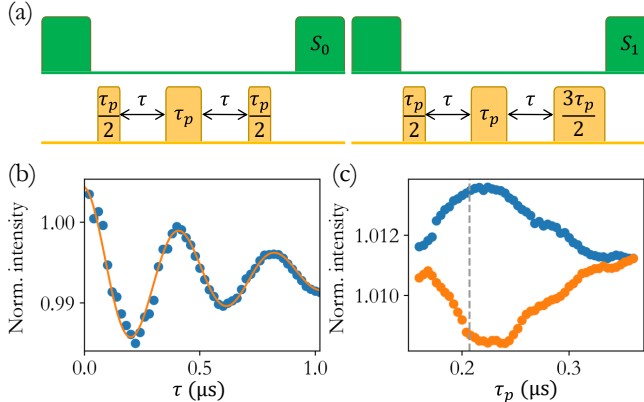

FIG. 14. (a) Hahn echo sequence used for optimization of $\pi$ and $\pi/2$-pulses. (b) Rabi oscillations with cosine fit (solid line). (c) Echo amplitude versus pulse duration $\tau_p$. The delay is $\tau = 500$ ns. Dashed line indicates the $\pi$-pulse duration extracted from fit of Rabi oscillations in (b) which appears not to be optimal.

cially available instruments and an open-source interface. In particular, we show that a relatively inexpensive acquisition card equipped with analog-to-digital converter, together with our upgrades on the Python-based software Qudi, allow anyone with minimal experimental skills to setup ODMR instrumentation capable of advanced magnetic resonance pulse sequences and address both individual optically-active spin-systems and large ensembles. We believe that our work will help foster the development of ODMR studies and their applications to sensing and metrology, quantum technologies, and material science, while also making ODRM methods more accessible to non-specialists. We encourage Qudi users to develop new instrument modules and make them available on GitHub to participate to wider openness and collaboration in research. We conclude by mentioning a few envisioned directions for both hardware and software developments in the coming years.

Performing ODMR at room temperature without space constraints does not present significant hardware challenges anymore. Yet, implementing similar measurements in confined environments, in ultra-high vacuum or at cryogenic temperatures, can rapidly become much more complex. While optical excitation and collection through free space or optical fiber is usually feasible in such circumstances, it can be challenging to locally provide enough MW power (without spurious heating) and apply strong enough and precisely oriented bias magnetic fields. Therefore, an integrated solution in the form of a chip-scale cryo-compatible device, with footprint below a few square millimeters, and which embeds a d.c. supplied MW oscillator and antenna, would be a valuable commodity for ODMR experiments. Such devices have already been developed in the context of conventional electron and nuclear spin resonance spectroscopy[149].

Regarding MW generation, voltage-controlled oscillators (VCOs) are inexpensive and compact alternatives to high-end MW sources, and lowering their phase noise would make them more useful for advanced measurements. Design of MW antennas delivering strong and homogeneous driving fields over wide areas is still in progress.

For single-emitter ODMR, both the digital single photon counting modules and the time tagging electronics drive the costs of the setup higher (typically 5'000 EUR each). As a cheaper alternative to the latter, FPGAs can be programmed for photon counting to yield much better temporal resolution than an NI card. An open-source package based on a commonly available FPGA for a do-it-yourself time tagger would make single emitter measurements more accessible.

For generating complex, long and low noise pulse sequences, NI cards reach their limits. As above, FPGA-based open-source solutions have the potential to make such experiments accessible at lower costs and with a high degree of customization for specific needs.

## ACKNOWLEDGMENTS

This project has received funding from the Swiss National Science Foundation (grants No. 185824, 170684, 198898), from the European Union's Horizon 2020 research and innovation programme under the Marie Skł odowska-Curie grant agreement No. 754354, and from EPFL Interdisciplinary Seed Fund.
We acknowledge Umut Yazlar from Appsilon B.V. (Delft, Netherlands) for his valuable contribution in providing single crystal diamond samples with desired characteristics.

## AUTHOR DECLARATIONS

### Conflict of Interest

The authors declare no conflicts of interest.

### Author Contributions

H.S and H.B. contributed equally to this work. H.B contributed significantly to the coding and assisted with the writing. H.S played a key role in writing the paper and valuably contributed to the coding. E.L contributed to the writing and revisions. H.B built the characterization setup for measurements. V.G. conducted, analyzed, and reported the measurements. M.C guided the writing throughout the project. M.C and C.G supervised the work and contributed to the writing and revisions.

## DATA AVAILABILITY

The original and updated version of qudi are available on github[130,132].

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
