# Peer review of "Optically detected magnetic resonance with an open source platform"

_SciPost Physics_

## Round 1 · Referee Report · Anonymous (Referee 1) · 2022-12-5

Report

This paper clearly describes ODMR measurements with NV centers and the experimental setup it implies. It presents very meaningful discussions on techniques enabling the acquisition of the data. It introduces the software Qudi designed on purpose and explains how one can adapt it to its own experimental setup.

It can be a resource of great interest for researchers who are about to develop an ODMR setup. However, some revisions are needed to improve the clarity of the paper and to make it more valuable.

I recommend the publication of the article in SciPost if the following points are adressed:

-Section I is an overview of the systems that allows ODMR . It would be nice to add a discussion on the peculiarites of NV center like systems which enable the experimental techniques described in the following section. For exemple, off resonant excitation is possible with NV centers and not with rare earth ions.

-Regarding section II, the authors should precise the minimum equipement required to perform ODMR (including the sample). It would be useful for teaching.

  • Table II stands that the light source needs a TEC. Is it really needed to realize the measurements described in section V? I would assume that cheaper Thorlabs lasers as CPS532 can be added to the list.

  • DS instruments MW sources (SG4400L, SG6000L) can also be added as cheap alternatives.

-In table II, time-taggers are mentionned. It is quite confusing because as far as I understand, the time tagging method described in the text do not rely on this kind of expensive devices.

-Section IV, is a good but brief introduction to Qudi and explains how to adapt it to its own experimental equipement. The paper should point to the online documentation http://ulm-iqo.github.io/qudi-generated-docs/html-docs/ . Is it up to date regarding the added features mentionned in the paper?

-Section V is of great interest for people mounting their own ODMR setup. For that purpose, the authors should indicate what is the used sample for the measurements of figs 2, 3, 12, 13, 14 and 15. They should also provide exemples of cheap diamonds that can be used for testing the setup before going to more specific samples or for student labworks. I suggest MSY diamond for exemple.

-Section V also lacks numbers in the text. What is the typical PL count rates and contrast one can expect to achieve? What are the T1 and the T2 tipically obtained?

-Is it possible to use QUDI for more complexe sequences such as dynamical decoupling? A small discussion about it should be added. Do you advise to use Qudi in order to control experiments involving more devices and complex sequences? For exemple, to study quantum memories based on rare earth ions that you mention in table I?

Attachment

---

## Round 2 · Referee Report · Anonymous · 2023-2-27

Report

The authors well adressed most of my comments and I recommend the publication of the article after very minor changes.

-Some arrows of Fig. 1 are not visible.

-Regarding Table 2, I think the authors should write the CPS 532 price which is less than 300 euros. In my opinion, this laser is enough to do very nice ODMR mesurements especially for pedagogical purpose.

-Table 2, what is the asterisk on "Light source" for?

---

## Round 2 · Author Response

Authors -- We thank the reviewer for his constructive comments and positive evaluation. We detail below our answers to each point raised and mention the change we made on the manuscript. The detail of the changes is given in the entry "List of changes"

Referee 0) This paper clearly describes ODMR measurements with NV centers and the experimental setup it implies. It presents very meaningful discussions on techniques enabling the acquisition of the data. It introduces the software Qudi designed on purpose and explains how one can adapt it to its own experimental setup. It can be a resource of great interest for researchers who are about to develop an ODMR setup. However, some revisions are needed to improve the clarity of the paper and to make it more valuable. I recommend the publication of the article in SciPost if the following points are addressed:

Referee 1) Section I is an overview of the systems that allows ODMR. It would be nice to add a discussion on the peculiarities of NV center like systems which enable the experimental techniques described in the following section. For example, off resonant excitation is possible with NV centers and not with rare earth ions.

Authors -- The NV center indeed gathers distinguishable peculiarities among other ODMR systems. The possibility to drive it off resonance, at room temperature is particularly important to to explain NV centers’ feature responsible for its popularity. It allows to perform ODMR with one “cheap” laser only and microwave equipment. Most of the other systems that we discuss, including rare earth ions, might require additional better lasers (narrow linewidth and tunable). However, while we take the NV center as an accessible example for ODMR, the equipment and method presented in the following sections are applicable to others as well which similarly require optical and microwave control. We updated the Sec. I.C and itemized the NV center peculiarities. We changed the introduction of Sec. II and the Table I to highlight the NV center’s peculiarities and the change one has to make to address other ODMR systems.

2) Regarding section II, the authors should precise the minimum equipment required to perform ODMR (including the sample). It would be useful for teaching.

-- In Sec II.A, we added a reference to the Figure 4 as a “minimal required optical setup to perform ODMR”. We updated the Table II to highlight (underline) the minimum equipment for ODMR,

3) Table II stands that the light source needs a TEC. Is it really needed to realize the measurements described in section V? I would assume that cheaper Thorlabs lasers as CPS532 can be added to the list.

-- Indeed, laser with temperature control using TEC may not be necessary in all experiments involving NV centers, and the specific requirements will depend on the experimental goals and the properties of the ODMR system being studied. However, fluctuations in laser power vastly affects the contrast of the ODMR, as well as the stability and coherence of the NV centers. It is therefore important to carefully control the laser power fluctuation against temperature in experiments involving NV centers to ensure that the properties of the NV centers which also depend on temperature are consistent and reproducible. We added that discussion in Sec. II.A.3.

4) DS instruments MW sources (SG4400L, SG6000L) can also be added as cheap alternatives.

-- We Added the suggested MW sources in table II

5) In table II, time-taggers are mentioned. It is quite confusing because as far as I understand, the time tagging method described in the text do not rely on this kind of expensive devices.

-- Indeed, the time tagging method we describe in Sec. III offers sufficient time resolution for most ODMR operations, including pulse measurements. However, when measuring the photoluminescence lifetime or autocorrelation function, faster devices can still be useful. They are often used by default in ODMR setup. We modified the related paragraph in Sec. II.A.4 to clarify the question.

6) Section IV, is a good but brief introduction to Qudi and explains how to adapt it to its own experimental equipement. The paper should point to the online documentation http://ulm-iqo.github.io/qudi-generated-docs/html-docs/ . Is it up to date regarding the added features mentionned in the paper?

-- This documentation both presents qudi’s general features which applies to our changes as well, and details the embedded classes based on the “in code” comments of the developers. As Qudi is an open source software, different users are actively participating in its development such that several versions forked from the initial one. While our version is already visible in the official github and directly accessible on ours. Some adjustments in our coding style and commenting are still required before we can send proper merge requests. If accepted this would automatically update the documentations with our own classes. Nonetheless our guithub will remain available with specific codes that are not to be generalized. We are now modifying our coding style accordingly. We rearranged the section IV’s introduction to better describe the reference to github and documentation’s urls adding the one suggested.

7) Section V is of great interest for people mounting their own ODMR setup. For that purpose, the authors should indicate what is the used sample for the measurements of figs 2, 3, 12, 13, 14 and 15. They should also provide examples of cheap diamonds that can be used for testing the setup before going to more specific samples or for student labworks. I suggest MSY diamond for exemple.

-- This indeed represents an important omission in our manuscript. We added the reference to the diamond we used in Sec.V and suggested a few diamond and nanodiamond suppliers in our Table II. It is worth noting that the cheapest diamonds may not always be the best choice for a given experiment, as the quality of the diamond can affect the properties of the NV centers. Therefore it might enforce experimental setup to have more precise and expensive equipment specifically in case of pulse measurements. We added this discussion in Sec. V introduction

8) Section V also lacks numbers in the text. What is the typical PL count rates and contrast one can expect to achieve? What are the T1 and the T2 tipically obtained?

-- There again those pieces of information were missing. We updated the Sec. V to include them.

9) Is it possible to use QUDI for more complexe sequences such as dynamical decoupling? A small discussion about it should be added. Do you advise to use Qudi in order to control experiments involving more devices and complex sequences? For example, to study quantum memories based on rare earth ions that you mention in table I?

-- With Qudi pulse measurement interface, it is possible to design more complex pulse sequences such as dynamical decoupling sequences via Qudi. Some of these protocols, like XY8 pulses are among the predefined methods and are available to choose via the graphical interface. New pulse sequences can easily be added to the program as explained in the documentation. If more than one control variable is desired, for example sweeping dark times between pulses and changing the duration of MW pulses, one can design and automate a series of independent experiments that sweep one variable while changing the other during each experiment using the Jupyter notebooks of Qudi We modified the Sec. IV.A to add those discussion.

As discussed in previous answers, any ODMR system would require similar optical pumping and microwave excitations. Although rare earth ion quantum memories would deserve specific care in the choice and control of laser wavelength, and/or dedicated pulse sequences, we do not see any reason to discard qudi for studying them. We believe this is now fairly described with the changes related to comment 1) we described above.

Authors -- We also took into account the the comment made in text and size the occasion to check the misspelling and typo and syntax all the text long. We also improuve the layout, using bullets when useful. We updated the affiliations the acknowledgment and the authors contributions section.

---

## Round 2 · List of Changes

1)
Sec. I.C:
For the following reasons, NV centers then became the most
studied ODMR system:
• ISC combined with a low Debye-Waller factor allows for an efficient off-resonance optical pumping and readout over a large range of wavelengths, even far over room temperature35 and under high pressures36.
• Diamond is bio-compatible making NV-doped nanodiamonds particularly relevant37 when used as biomarkers38–40 or quantum sensors41–43
• NV centers can exhibit long spin relaxation and coherence times17 thanks to the low density of nuclear spins in diamond and the decoupling of spin states from lattice phonons16. It makes them versatile quantum sensors44 for detection and imaging of magnetic and electric fields, temperature, strain, currents and associated noises9,45,46, as well as for microwave field imaging47 or spectroscopy48,49
• NV center ODMR properties can be extended to polarize50–52 and/or detect53–56 other electron or nuclear spins in their vicinity and perform their magnetic resonance spectroscopy57,58. In these cases the sensitivity can go down to a single nuclear spin, orders of magnitude better than conventional electron paramagnetic resonance (EPR) and nuclear magnetic resonance (NMR) methods.
• Finally, thanks to their excellent coherence properties and couplings to even longer lived nuclear spins59, NV centers are also promising qubits for quantum information processing11 and fundamental studies of quantum mechanics60

Sec.II, introduction:
In this section we review the instrumental requirements for ODMR measurements in terms of optical setup, MW generation and biased magnetic field. Both CW and pulsed ODMR measurements are discussed. We take the diamond NV center as an accessible example (green pumping, red photoluminescence (PL)). However, the properties of other defects summarized in Table I can be consulted in order to adapt the equipment for their study, with a special care given on the optical wavelengths, microwave frequencies and possible need for cryogenic environment.

Table I: see manuscript.

2)
Sec. II.A
The simple optical setup reported in Fig.4, analogous to an epifluorescence microscope can represent the minimal required optical setup to perform ODMR.

Table I: see manuscript.

3)
Table II: We removed TEC from table II and added Thorlabs lasers CPS 532

Sec. II.A.3.
However, fluctuations in laser power affects the contrast of the ODMR, as well as the stability and coherence of the NV centers. Therefore, it is important to carefully control the laser power fluctuation against temperature.

4)
We Added the suggested MW sources in table II

5) Sec. II.A.4
Different approaches are used to measure the PL as a function of the type of signal coming from the photodetector.
Analog photodetectors (a-PDs) [...]
Digital photocounters [...]. Such resolution can be useful to resolve the PL lifetime (see Sec. V or to acquire second order autocorrelation function g^{(2)} (e.g. to evidence single photon operations98-100. Otherwise, ODMR rarely involves that fast processes. Such a time precision can then become a disadvantage as the instrument buffer becomes more rapidly saturated by the recorded time tags.
A field-programmable gate array (FPGA) can be used as a cost effective alternative to time-tagging instrument, but it requires dedicated programming skills.
DAQ cards also include general purpose timers/counters. As presented in Sec. III A it is possible to configure them for photon counting with sampling rate up to twice their internal clock, which is two orders of magnitude faster than their analog inputs.

6)
end of Sec. IV
The official
Qudi’s version is available on Github130 and a general documentation based on evelopers "in code" comments along with a brief general tutorial can be found here131. When not
already merged within the official Github, our added features can be found on our Github page forked from it132

7)
Sec. V introduction
The measurements presented in this section are performed on a bulk diamond sample enriched in NV centers and provided by Appsilon B.V. Other possibilities for commercial diamond providers are summarized in Table II. It should be noted that the figures of merit discussed here can vary significantly depending on the material quality. [For practical purposes, we provide throughout the section typical values for these figures of merit in an NV ensemble. They are given for a sample at room temperature, in the absence of external perturbation (magnetic field, electric field, pressure, etc.).]

8) Sec. V introduction
[It should be noted that the figures of merit discussed here can vary significantly depending on the material quality.] For practical purposes, we provide throughout the section typical values for these figures of merit in an NV ensemble. They are given for a sample at room temperature, in the absence of external perturbation (magnetic field, electric field, pressure, etc.).

Our changes regarding the photon count ratein Sec. V.A:
A single NV center can display photon emission rates of up to few 100kHz with a microscope numerical aperture exceeding 0.9 and a laser excitation power in the mW range. The total PL depends linearly on the number of NVs in the ensemble. According to the level of brightness of the sample and collection efficiency of the setup, [...]
Our change regarding contrast in Sec. V.B:
Typically, a single NV center can exhibit an ODMR contrast above $10\%$. For an ensemble with 4 possible NV orientations, this results in best contrast of around $2\%$ for each resonance in the ODMR spectrum, close to the values reported in Fig.~12. The presence of other photoluminsecent defects, such as neutral NV0, causes a reduction of the ODMR contrast.

Our changes regarding T1 & T2 in Sec. V.C:
As stated earlier, T1, T2 and T2* depend greatly on the diamond quality, as well as on the presence of external perturbations. For an NV ensemble with moderate concentration (below 1ppm), T1 is usually on the order of several milliseconds, T* ~ 0.5-1 us and T2 is limited to few 10s of us with a simple spin-echo sequence, but can be extended to several 100 us with better dynamical decoupling pulse protocols. The interested reader should refer to e.g.43 for a more quantitative, in-depth discussion of these coherence times, which is beyond the scope of the present paper.

[43] J. F. Barry, J. M. Schloss, E. Bauch, M. J. Turner, C. A. Hart, L. M. Pham, and R. L. Walsworth, “Sensitivity optimization for NV-diamond magnetometry,” Reviews of Modern Physics 92, 15004 (2020)

9)
In Sec. IV.A:
Besides the aforementioned pulse protocols, it is possible to design more complex pulse sequences such as dynamical decoupling sequences via Qudi. Some of these protocols, like XY8 pulses are among the predefined methods and are available to choose via the graphical interface. New pulse sequences can easily be added to the program as explained here[135] .
Several tools are available to extract and analyze the pulses on the go and to automatize the setting of experimental parameters, which facilitates long measurement sequences across many different material systems. For instance if it is necessary to vary than one control variable, (e.g. sweeping dark times between MW pulses and their duration), one can use qudi’s Jupiter notebooks[136] to design and automate a series of independent experiments in which one variable is swept repeatedly while the second is changed step by step in between.

AFFILIATIONS)
Hossein Babashah,1, 2, a) Hoda Shirzad,1, 3, a) Elena Losero,1, b) Valentin Goblot,1, 3 Christophe Galland,1, 3 and
Mayeul Chipaux1, 3, c)
1)Institute of Physics, École Polytechnique Fédérale de Lausanne (EPFL), Lausanne CH-1015, Switzerland
2)WAINVAM-E, 1 rue Galilée, 56270 Plœmeur, France
3) Center for Quantum Science and Engineering, EPFL, Lausanne, Switzerland

ACKNOWLEDGMENTS)
This project has received funding from the Swiss National Science Foundation (grants No. 185824, 170684, 198898), from the European Union's Horizon 2020 research and innovation programme under the Marie Skł odowska-Curie grant agreement No. 754354, and from EPFL Interdisciplinary Seed Fund.
We acknowledge Umut Yazlar from Appsilon B.V. (Delft, Netherlands) for his valuable contribution in providing single crystal diamond samples with desired characteristics.

AUTHOR CONTRIBUTIONS)
H.S and H.B. contributed equally to this work. H.B contributed significantly to the coding and assisted with the writing. H.S played a key role in writing the paper and valuably contributed to the coding. E.L contributed to the writing andrevisions. H.B built the characterization setup for measurements. V.G. conducted, analyzed, and reported the measurements. M.C guided the writing throughout the project. M.C and C.G supervised the work and contributed to the writing and revisions.

General TYPO/ PAYOUT)
See all along the text

You are currently on this page

Resubmission 2205.00005v2 on 9 January 2023

---

## Editorial Decision

resubmitted